# An Adaptive Orthogonal Convolution Scheme for Efficient and Flexible CNN Architectures

Thibaut Boissin [1] [2]   Franck Mamalet [1]   Thomas Fel [3]   Agustin Martin Picard [1]   Thomas Massena [4] [2]
Mathieu Serrurier [2]

## Abstract

Orthogonal convolutional layers are valuable components in multiple areas of machine learning, such as adversarial robustness, normalizing flows, GANs, and Lipschitz-constrained models. Their ability to preserve norms and ensure stable gradient propagation makes them valuable for a large range of problems. Despite their promise, the deployment of orthogonal convolution in large-scale applications is a significant challenge due to computational overhead and limited support for modern features like strides, dilations, group convolutions, and transposed convolutions. In this paper, we introduce **AOC** (Adaptive Orthogonal Convolution), a scalable method that extends the work of (Li et al., 2019), effectively overcoming existing limitations in the construction of orthogonal convolutions. This advancement unlocks the construction of architectures that were previously considered impractical. We demonstrate through our experiments that our method produces expressive models that become increasingly efficient as they scale. To foster further advancement, we provide an open-source python package implementing this method, called Orthogonium.

## 1. Introduction and Related Works

Orthogonal layers have become fundamental components in various deep learning architectures due to their unique mathematical properties, which offer benefits across multiple applications. For instance, robustness against adversarial attacks can be achieved by managing a model's Lipschitz constant (Szegedy et al., 2014) – with 1-Lipschitz networks

being a prime candidate: such networks allowing the computation of robustness certificates. Initially, researchers experimented with regularization techniques (Cisse et al., 2017); however, constrained networks, especially those employing orthogonal layers, soon became central, as they provided the advantage of tighter certification bounds: The overall Lipschitz constant of a sequence of layers is typically estimated as the product of the individual layer constants; however, this bound is often loose, and computing the exact constant is NP-hard (Virmaux & Scaman, 2018). (Anil et al., 2019) shows that combining MaxMin activation with orthogonal layers makes the aforementioned bound tight. Beyond robustness, orthogonal layers also play a key role in enhancing performance in normalizing flows. Normalizing flows are generative models that transform simple distributions into complex ones via invertible mappings (Dinh et al., 2017; Rezende & Mohamed, 2015). Orthogonal convolutions enable these transformations with a computable Jacobian determinant, thus improving training efficiency (Kingma & Dhariwal, 2018) and forming the basis for invertible residual networks (Behrmann et al., 2019). Additionally, orthogonal layers stabilize training deep and recurrent neural networks (RNNs) by preserving gradient norms through time, essential in capturing long-term dependencies in time-series, such as language and speech tasks (Kiani et al., 2022; Qi et al., 2020; Bansal et al., 2018). Lastly, in Wasserstein GANs (WGANs) (Arjovsky et al., 2017) orthogonality in both the discriminator and generator (Müller et al., 2019) supports stability and expressivity without requiring weight clipping or gradient penalties (Gulrajani et al., 2017), making it essential for large-scale GAN training (Brock et al., 2018; Miyato et al., 2018). See Appendix A for a detailed list of applications.

Despite these benefits, extending orthogonality to convolutional layers remains challenging. The orthogonalization of large Toeplitz matrices-structures central to convolution- is challenging to do without compromising its convolutional properties and is sometimes non-feasible (Achour et al., 2022). Early approaches (Wang et al., 2020; Qi et al., 2020) explored regularization, yet practical constraints led to the following solutions:

**Explicit Construction Methods.** Building on (Xiao et al.,

[1]Institut de Recherche Technologique Saint-Exupery, France [2]IRIT, France [3]Kempner Institute, Harvard University, USA [4]Innovation & Research Division, SNCF, France. Correspondence to: Thibaut Boissin <thibaut.boissin@irt-saintexupery.com>, Franck Mamalet <franck.mamalet@irt-saintexupery.com>.

*Proceedings of the $42^{nd}$ International Conference on Machine Learning*, Vancouver, Canada. PMLR 267, 2025. Copyright 2025 by the author(s).

| Method | | Orthogonal | Equivalent Kernel Size | Code available | Change Channels | Stride | Conv Transpose | Groups | Dilation |
|---|---|---|---|---|---|---|---|---|---|
| BCOP | (Li et al., 2019) | ✓ | $k$ | ✓ | ✓ | ≈ | ✗ | ✗ | ✗ |
| SC-FAC | (Su et al., 2022) | ✓ | $k$ separable | ✗ | ✓ | ✓ | ≈ | ✓ | ✓ |
| ECO | (Yu et al., 2022) | ✓ | $I_w \times I_h$ | ✗ | ≈ | ≈ | ✗ | ✗ | ≈ |
| Cayley | (Trockman & Kolter, 2021) | ✓ | $I_w \times I_h$ | ✓ | ≈ | ≈ | ≈ | ✗ | ✗ |
| LOT | (Xu et al., 2022) | ✓ | $I_w \times I_h$ | ✓ | ✓ | ≈ | ✗ | ✗ | ✗ |
| ProjUNN-T | (Kiani et al., 2022) | ✓ | $I_w \times I_h$ | ✓ | ✗ | ✗ | ✗ | ✗ | ✗ |
| SLL | (Araujo et al., 2023) | ✗ | composed | ✓ | ✗ | ✗ | ✗ | ✗ | ✗ |
| Sandwich | (Wang & Manchester, 2023) | ✗ | composed | ✓ | ≈ | ≈ | ✗ | ✗ | ✗ |
| AOL | (Prach & Lampert, 2022) | ✗ | $k$ | ✓ | ✓ | ✓ | ✗ | ✗ | ✗ |
| SOC | (Singla & Feizi, 2021b) | ✓ | $k + \left(n\frac{k}{2}\right)$ | ✓ | ≈ | ≈ | ✗ | ✗ | ✗ |
| **AOC** | **(Ours)** | ✓ | $k$ | ✓ | ✓ | ✓ | ✓ | ✓ | ✓ |

Table 1: **Comparison of orthogonal convolution methods.** A check mark (✓) indicates full support for the feature, a cross mark (✗) indicates lack of support (in the implementation), and an approximate symbol (≈) indicates partial support (emulation). Here, $k$ denotes the kernel size, and $I_w \times I_h$ represents the input dimensions. The pip package of AOC provides an implementation supporting strides, dilations, group convolution and transposed convolution. This work also covers the conditions under which other methods could support those features.

2018), approaches like BCOP (Li et al., 2019), SC-Fac (Su et al., 2022), and ECO (Yu et al., 2022) construct orthogonal convolutions directly in the spatial domain. These methods maintain orthogonality but often lack flexibility in kernel size control and do not support operations like striding and transposed convolutions.

**Frequency Domain Approaches.** Methods such as Cayley Convolution (Trockman & Kolter, 2021), LOT (Xu et al., 2022), and ProjUNN-T (Kiani et al., 2022) enforce orthogonality by parameterizing kernels in Fourier space, albeit at the cost of increased computational complexity and constraints on spatial or grouped convolutions.

**Composite Layer Techniques.** Skew Orthogonal Convolutions (Singla & Feizi, 2021b) combine multiple convolutional layers to approximate orthogonality, resulting in an orthogonal block that uses convolutions.

**Mitigating vanishing gradients.** In some cases, strict orthogonality is relaxed. Leading to layers that are 1-Lipschitz while mitigating vanishing gradients. This avoids the computational demands of full orthogonalization (Prach & Lampert, 2022; Meunier et al., 2022; Araujo et al., 2023).

Despite their success in certifiable adversarial robustness, these methods have seen limited adoption in other fields. This is largely due to implementation limitations: current layers fail to support essential features like stride, groups, or dilation, which are critical for architectures such as U-Nets (Ronneberger et al., 2015), GANs (Goodfellow et al., 2014), VAEs (Kingma, 2013) (upsampling convolutions), dilated CNNs (Li et al., 2018) (dilation), and modern models like ResNeXt (Xie et al., 2017) and EfficientNet (Tan &

Le, 2019) (grouped convolutions). On the other end, performance in certifiable adversarial robustness is tied to model size and training duration (Prach et al., 2024). However, 1-Lipschitz constrained networks lack the usual scaling laws (Prach & Lampert, 2024), often requiring massive datasets even for small-scale problems (Hu et al., 2023). Developing an efficient and scalable method that supports modern features method could address both challenges simultaneously.

**Our Contributions.** In response to the limitations of existing methods, we introduce **A**daptive **O**rthogonal **C**onvolution (**AOC**), a method that combines two existing approaches to construct convolution layers that address the following key constraints *simultaneously* while remaining efficient:

- **Orthogonal**: AOC enforces strict orthogonality, allowing convolutional layers to retain essential properties across applications.

- **Explicit**: In contrast to frequency domain methods, AOC generates explicit convolution kernels in the spatial domain, allowing straightforward implementation in standard deep learning frameworks without specialized operations or significant computational overhead.

- **Flexible**: Supporting a range of essential operations – including striding, transposed convolutions for upsampling, grouped convolutions, and dilation – AOC adapts effectively to modern neural network architectures.

- **Scalable**: Designed for large-scale applications, our

implementation maintains efficiency, incurring only a 10% slowdown compared to unconstrained models in realistic ImageNet 1K (Deng et al., 2009) training setup.

To underscore the advantages of our method, we include a comparative summary in Table 1, highlighting support for key features across different methods. By combining orthogonality, explicit construction, and flexibility, our approach seamlessly bridges theoretical rigor with practical efficiency in deep learning models. Beyond AOC, this work introduces a mathematical framework that unlocks improvements of other existing methods such as SOC, SLL, or Sandwich (See Appendix E). Our implementation was rigorously tested and is available in a package called Orthogonium, which also contains improved implementations of other existing methods.

The paper is organized as follows: Section 2 outlines the three main aspects of AOC – its core tools, kernel construction, and scalable implementation. Section 3.2 presents an evaluation of the method's performance in terms of speed and expressive power. Finally, Section 4 discusses how our approach can enhance existing methods in the literature.

## 2. An Adaptive scheme to build Orthogonal Convolution (AOC)

We will first recall the definition and properties of the Block-convolution in Section 2.1. This tool is used in Section 2.2 to construct orthogonal kernels explicitly, with the support of modern convolution features. Finally, Section 2.3 provides implementation details that allow the method to scale.

### 2.1. Core tool: Block-convolution

Our approach builds upon three foundational papers: (Xiao et al., 2018), which generalized orthogonal initialization to convolution to enable training networks with 10 000 layers, though without addressing constrained training; (Su et al., 2022), which tackled this for separable convolutions; and (Li et al., 2019), which extended it to general 2D convolutions. These works rely heavily on a tool known as Block-convolution. In this section, we review, clarify, and extend this mathematical framework.

**Notations:** We consider convolutional layers characterized by $c_o$, the number of output channels; $c_i$, the number of input channels; $k_1 \times k_2$, the kernel size; $s$, the stride parameter; $g$, the number of groups; $d$ the dilation rate. For simplicity in the notation, we fix the group parameter to $g = 1$ and $d = 1$ by default (all proofs hold for other values of $g$ and $d$, see Section 2.2). Also, we assume circular padding in all proofs. The kernel tensor of the convolution is denoted $\mathbf{K} \in \mathbb{R}^{c_o \times c_i \times k_1 \times k_2}$, while $x \in \mathbb{R}^{c_i \times h \times w}$ denotes

the input tensor. We describe the convolution operation with three equivalent notations:

$$y = \mathbf{K} \star_s x \quad \text{(Kernel notation)} \quad (1)$$

$$\bar{y} = \mathcal{S}_s \mathcal{K} \bar{x} \quad \text{(Toeplitz notation)} \quad (2)$$

$$y = \text{conv}_K(x, \texttt{stride} = s) \quad \text{(Code notation)} \quad (3)$$

Equation (1) defines the convolution operation with kernel $\mathbf{K}$ and stride $s$ applied on $x$. Equation (2) highlights that this convolution is equivalent to a linear operation defined by a matrix product between a Toeplitz matrix $\mathcal{K} \in \mathbb{R}^{c_o h w \times c_i h w}$ and a vector $\bar{x} \in \mathbb{R}^{c_i h w}$, which is obtained by flattening $x$. The striding operation is represented by a masking diagonal matrix $\mathcal{S}_s \in \mathbb{R}^{c_o \frac{h}{s} \frac{w}{s} \times c_o h w}$ with ones on the selected entries and zeros elsewhere. When $s = 1$, we have $\mathcal{S}_1 = \mathcal{I}$, the identity matrix (with kernel $\mathbf{I}$). Equation (3) shows these notations in pseudo-code form.

**Definition 2.1** (Block-convolution $\circledast$[1]). *The Block-convolution, denoted as $\mathbf{B} \circledast \mathbf{A}$, computes the equivalent kernel of the composition of two convolutional kernels, $\mathbf{A}$ and $\mathbf{B}$, enabling their combined effect without performing each convolution separately.*

$$(\mathbf{B} \circledast \mathbf{A}) \star_s x = \mathbf{B} \star_s (\mathbf{A} \star_1 x) \quad (4)$$

$$\mathcal{S}_s(\mathcal{B}\mathcal{A})\bar{x} = (\mathcal{S}_s \mathcal{B})\mathcal{A}\bar{x} \quad (5)$$

$$\text{conv}_{B \circledast A}(x, s) = \text{conv}_B(\text{conv}_A(x, 1), s) \quad (6)$$

This operator assumes that the number of input channels of the second convolution $\mathbf{B}$ matches the number of output channels of the first convolution $\mathbf{A}$ (condition denoted as $A \bowtie B$). Given $\mathbf{A} \in \mathbb{R}^{c_{int} \times c_i \times k_1^A \times k_2^A}$ and $\mathbf{B} \in \mathbb{R}^{c_o \times c_{int} \times k_1^B \times k_2^B}$, then $\mathbf{B} \circledast \mathbf{A} \in \mathbb{R}^{c_o \times c_i \times (k_1^A + k_1^B - 1) \times (k_2^A + k_2^B - 1)}$. The computation of Block-convolution kernel weights is given by:

$$(\mathbf{B} \circledast \mathbf{A})_{m,n,i,j} = \sum_{c=0}^{c_{int}-1} \sum_{i'=0}^{k_1^B-1} \sum_{j'=0}^{k_2^B-1} \mathbf{B}_{m,c,i',j'} \cdot \mathbf{A}_{c,n,i-i',j-j'}$$

$$(7)$$

where $\mathbf{A}$ is zero-padded, i.e., $\mathbf{A}_{c,n,i,j} = 0$ if $i \notin [0, k_1^A[$ or $j \notin [0, k_2^A[$. While a classic result, for completeness, we provide the proof for 1D convolution kernels in Appendix G.1. This operator $\circledast$, using matrix products between order 4 tensors, should not be confused with standard convolution, which takes a 4-dimensional tensor and a 3-dimensional input tensor. The complexity of these weight computations is $\mathcal{O}(c_o c_i c_{int} \prod_{p=1}^{2} (k_p^A + k_p^B - 1) k_p^B)$, necessitating an efficient implementation, detailed in Section 2.3.

**Block-convolution and Orthogonality:** We will recall the definition of an orthogonal convolution, using Eq. (5) and the definition of an orthogonal matrix:

---

[1]Initially denoted by (Li et al., 2019) as □

**Definition 2.2** (Orthogonal Convolution). *A convolution defined by Eq.* (2) *is row/column orthogonal iff:*

$$(\mathcal{S}_s\mathcal{K})(\mathcal{S}_s\mathcal{K})^T = \mathcal{I} \quad \text{(row orthogonal)}$$
$$(\mathcal{S}_s\mathcal{K})^T(\mathcal{S}_s\mathcal{K}) = \mathcal{I} \quad \text{(column orthogonal)}$$

The type of orthogonality (row or column) is determined by the larger dimension of the matrix in the toeplitz notation (Eq. (2)). When $c_i s^2 > c_o$, $\mathcal{S}_s\mathcal{K}$ is column orthogonal (Achour et al., 2022). When $c_i s^2 = c_o$, $\mathcal{S}_s\mathcal{K}$ is a square matrix, and the two conditions are equivalent. Unless explicitly stated, we will use the term orthogonal to refer to row (resp. column) orthogonal. Finally, when $s = 1$, the condition on the Toeplitz matrices is equivalent to $\mathbf{K} \circledast \mathbf{K}^T = \mathbf{I}$ (resp. $\mathbf{K}^T \circledast \mathbf{K} = \mathbf{I}$). A formal definition of $\mathbf{K}^T$ can be found in Definition 2.6.

Through this paper, we leverage the Block-convolution property to compute implicitly the composition of convolutions. This operation preserves orthogonality when the following requirement is met:

**Proposition 2.3** (Composition of Orthogonal Convolutions). *The composition of two row orthogonal convolutions is a row orthogonal convolution:*

$$\mathcal{A}\mathcal{A}^T = \mathcal{I} \quad and \quad \mathcal{B}\mathcal{B}^T = \mathcal{I} \implies \mathcal{A}\mathcal{B}(\mathcal{A}\mathcal{B})^T = \mathcal{I}$$

*The same applies to two column orthogonal convolutions.*

The proof is straightforward using the Toeplitz notation and matrix multiplications. Note that, the composition of a row orthogonal with a column orthogonal convolution is, in general, not orthogonal. This will significantly influence the construction of AOC convolutions.

### 2.2. Construction of Strided, Transposed, Grouped, Dilated, Orthogonal Convolutions with AOC

As mentioned previously, existing methods for explicit orthogonal convolution construction face several challenges that hinder their scalability in modern CNNs. Frequency-domain approaches, for instance, require computing the FFT on input images, which becomes prohibitive for large image resolutions (e.g., 224×224 in ImageNet-1K). Methods like SOC, LOT, ECO, and Cayley (Singla & Feizi, 2021b; Xu et al., 2022; Yu et al., 2022; Trockman & Kolter, 2021) handle cases where $c_i \neq c_o$ through inefficient padding or channel dropping, further limiting their practicality. Most of these methods also emulate striding, using a channel reshaping method called InvertibleDownSampling, significantly increasing the computational cost compared to native striding (see Section 2.2). Beyond inefficiency, these limitations prevent the adoption of essential modern convolutional features such as strides, transposition, groups, and dilation. Since these features were specifically designed to enhance

the efficiency of modern neural network architectures, emulation is not a viable option for computationally demanding scenarios.

**The strategy of AOC.** The core idea of our method is to combine several convolution kernels with an efficient implementation of the Block-convolution $\circledast$ to build any type of orthogonal convolution. We mainly build upon two type of kernels: one that guarantees orthogonality for any kernel size (when $s = 1$) and another that guarantees orthogonality for any stride $s$. BCOP and SC-Fac (Li et al., 2019; Su et al., 2022), by leveraging Block-convolution $\circledast$, are effective for the first kernel, whereas other methods lack control over the kernel size. For the second kernel, RKO (Serrurier et al., 2021) is the only viable option, as all other methods depend on stride emulation.

**Standard Orthogonal Convolution** For standard orthogonal convolutions with a fixed kernel size, we rely on three main works (Xiao et al., 2018), (Li et al., 2019), and (Su et al., 2022). In Appendix B, we unify these works within a consistent notation framework, highlighting similarities and differences among their approaches to constructing standard orthogonal convolutions (i.e., without stride, transposition, grouping, or dilation). These methods primarily rely on constructing elementary blocks ($1 \times 1$, $1 \times 2$, and $2 \times 1$ orthogonal convolutions) and assembling these blocks to create orthogonal convolutions of the desired size and shape. Briefly, both methods construct symmetric projectors (matrices such that $N = N^2 = N^T$ and $(I - N) = (I - N)^2 = (I - N)^T$). This property allows to construct orthogonal convolution whose kernel size is $1 \times 2$ (resp. $2 \times 1$).

(Li et al., 2019; Xiao et al., 2018) chose to compose $(k_1 - 1)$ $2 \times 1$ kernels $\mathbf{P_i}$ with $(k_2 - 1)$ $1 \times 2$ kernels $\mathbf{Q_i}$ and a $1 \times 1$ kernel $\mathbf{M}$ to form a $k_1 \times k_2$ kernel:

$$\mathbf{K}_{\text{BCOP}} = \underbrace{(\mathbf{P_{k-1}} \circledast \mathbf{Q_{k-1}})}_{\text{pairs of 1x2 and 2x1 kernels}} \circledast \ldots \circledast (\mathbf{P_1} \circledast \mathbf{Q_1}) \circledast \underbrace{\mathbf{M}}_{\text{1x1}}$$

A detailed description of the BCOP method is provided in Fig. 1, with additional details in Appendices B and G.3 for the proof of orthogonality. It is also worth noting that both approaches have the same number of parameters, which is lower than the number of parameters required to span all orthogonal convolutions. This is further discussed in Appendix F.

**Native Striding for Orthogonal Convolutions** Classical convolutional networks use strided convolutions. However, most existing work on orthogonal convolutions does not support stride directly, instead emulating striding via a reshaping operation (Shi et al., 2016): transforming the $c_i \times I_h \times I_w$ tensor into a $c_i s^2 \times \frac{I_h}{s} \times \frac{I_w}{s}$ tensor, followed by a non-strided

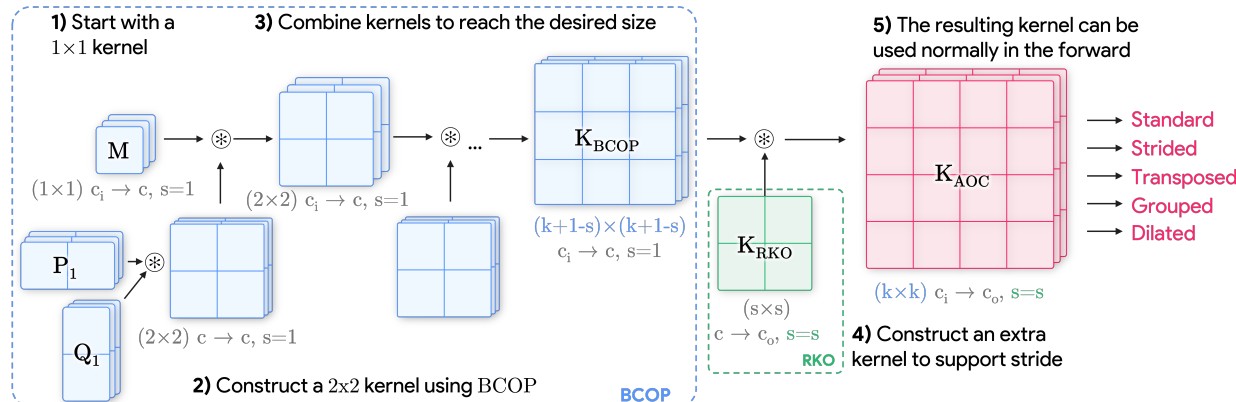

Figure 1: **AOC enables the construction of orthogonal kernels with customizable sizes and strides.** By leveraging the ⊛ operator, we can fuse kernels obtained from two existing methods, namely BCOP and RKO. With this approach, we can build orthogonal kernels that support native striding, effectively mitigating the drawbacks of the two base methods.

convolution. Unlike native striding, emulation increases the number of parameters of the convolution ($s^2$ times more parameters), which means constraining larger matrices. Since algorithms used for this tasks have cubic time complexity in the number of channels (Prach et al., 2024), emulation induces a prohibitive cost for strided convolutions ($s^6$ times slower). Beyond this limitation, emulated striding also prevents native implementation of transposed convolutions with stride (see Section 2.2). In this section, we propose a method to construct orthogonal kernels that support native striding.

To our knowledge, only two works (Serrurier et al., 2021; 2024) claim to use native stride. These rely on a method referred to by (Li et al., 2019) as Reshaped Kernel Orthogonalization (RKO). This method involves reshaping the kernel $\mathbf{K}_{RKO} \in \mathbb{R}^{c_o \times c_i \times k_1 \times k_2}$ into a matrix $K' \in \mathbb{R}^{c_o \times c_i k_1 k_2}$ and orthogonalizing it. In general, with the adequate multiplicative factor, the resulting convolution is 1-Lipschitz but not orthogonal. In this work, we prove that no additional factor is required when $k = s$ to obtain an orthogonal convolution. This result can be carefully combined with BCOP (Section 2.2) to provide orthogonal convolutions of desired kernel size and stride.

**Proposition 2.4** (RKO gives an orthogonal kernel when $k_1 = k_2 = s$)**.** *When $K' \in \mathbb{R}^{c_o \times c_i kk}$ is orthogonal, the convolution with the reshaped kernel $\mathbf{K}_{RKO} \in \mathbb{R}^{c_o \times c_i \times k \times k}$ and a stride $s = k$ is orthogonal.*

The proof of Proposition 2.4 relies on the introduction of a permutation matrix on the inputs, and is given in Appendix G.4. The proposed method, called AOC, combines the BCOP and RKO methods to construct a Strided Convolution with Arbitrary Kernel Size:

$$\mathbf{K}_{AOC} = \mathbf{K}_{RKO} \circledast \mathbf{K}_{BCOP} \qquad (8)$$

As shown in Fig. 1, by choosing specific sizes for $\mathbf{K}_{BCOP} \in \mathbb{R}^{c \times c_i \times (k_1+1-s) \times (k_2+1-s)}$ and $\mathbf{K}_{RKO} \in \mathbb{R}^{c_o \times c \times s \times s}$, the fusion results in an orthogonal convolution of kernel size $k_1 \times k_2$ and stride $s$.

While the formulation (Eq. (8)) may seem straightforward, the order of composition and the choice of the internal channel size $c$ are crucial to preserve orthogonality.

**Proposition 2.5** (Orthogonality of strided AOC)**.** *The composition of two kernels $\mathbf{K}_{BCOP} \in \mathbb{R}^{c \times c_i \times (k_1+1-s) \times (k_2+1-s)}$ and $\mathbf{K}_{RKO} \in \mathbb{R}^{c_o \times c \times s \times s}$ (Eq. (8)) with an internal channel size $c = max(c_i, \lfloor \frac{c_o}{s^2} \rfloor)$ yields an orthogonal convolution with stride.*

The proof of Proposition 2.5, relies on Proposition 2.3 which holds for strided convolutions applied to $\mathcal{S}_s \mathcal{K}_{RKO}$, is detailed in Appendix G.5 . We prove that when $min(c_i, \frac{c_o}{s^2}) \leq c \leq max(c_i, \frac{c_o}{s^2})$, the two matrices $\mathcal{K}_{BCOP}$ and $\mathcal{S}_s \mathcal{K}_{RKO}$ are either both row-orthogonal or both column-orthogonal. The choice of $c = max(c_i, \lfloor \frac{c_o}{s^2} \rfloor)$ maximizes the expressiveness of the parametrization while ensuring that the resulting convolution is orthogonal. Note that the size of the $\mathbf{K}_{BCOP}$ imposes the condition $k + 1 - s \geq 0$. However, according to (Achour et al., 2022), no orthogonal kernel exists when $s > k$. Therefore, our approach encompasses all valid configurations of orthogonal convolutions.

**Native Transposed Orthogonal Convolutions** In addition to the practical verification of orthogonality in Definition 2.2, transposed convolutions are mostly used as learnable upscaling layers in architectures such as U-Net (Ronneberger et al., 2015) or VAEs (Kingma, 2013; Van Den Oord et al., 2017). Even though some methods in Table 1 can provide transposition for standard convolution, none are applicable to strided convolutions due to their use of emulation via reshaping, as described in Section 2.2. Thus, AOC is the first method to support transposed orthogonal convolutions with upscaling.

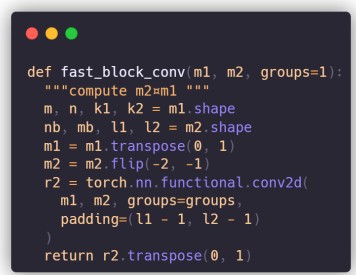

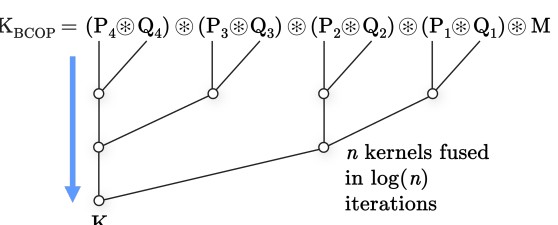

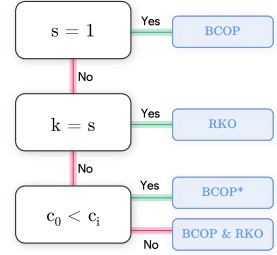

(a) **Fast block convolution.** We optimized the 2D convolution in order to compute the ⊛ operator with maximum parallelism.

(b) **Parallelize BCOP iterations.** By leveraging associativity of ⊛, we can compute the $n$ iteration in $\mathcal{O}\log(n)$ steps using parallel associative scan.

(c) **Parsimonious parametrization.** The method can sometimes simplify to quicker equivalent parametrization.

Figure 2: **Fast implementation of AOC.** We achieve a highly scalable parametrization thanks to optimizations at every level of our method: starting from the ⊛ operator 2a, to BCOP 2b, to our complete method 2c. It results in a method with a lower overhead as scale increases.

For a given convolution with stride defined by Eq. (2), the transposed convolution corresponds to the application of the transposed matrix $(\mathcal{S}_s\mathcal{K})^T$, inverting the role of $c_i$ and $c_o$. The resulting operation can be defined by the three notations:

**Definition 2.6** (Transposed Convolution). *A transposed convolution is defined as follows:*

$$y = \mathbf{K}^T \star_{\frac{1}{s}} x \tag{9}$$

$$\bar{y} = (\mathcal{S}_s\mathcal{K})^T \bar{x} = \mathcal{K}^T \mathcal{S}_s^T \bar{x} \tag{10}$$

$$y = \texttt{ConvTranspose}_K(x, \texttt{stride} = s) \tag{11}$$

The code notation (Eq. (11)) corresponds to the implementation in PyTorch parametrized by the original kernel $\mathbf{K}$. The Eq. (10) corresponds to the transposition of the underlying Toeplitz matrix. The kernel notation (Eq. (9)) can be viewed as a standard convolution with a transposed kernel and fractional striding. The kernel $\mathbf{K}^T \in \mathbb{R}^{c_i \times c_o \times k_1 \times k_2}$ is obtained by transposing the channel dimensions (the first ones) and reversing the kernel ones (the last two). By using an orthogonal direct convolution $\mathbf{K}_{AOC}$, this construction results in an orthogonal transposed convolution. Although the definition and proof are straightforward, the practical application requires the explicit construction of a strided orthogonal convolution kernel (as detailed in Eq. (8)).

**Proposition 2.7** (Transposed Orthogonal Convolution). *The transposition of a row orthogonal convolution is a column orthogonal convolution, and vice versa.*

The proof is straightforward with the Toeplitz matrix notation, but is given for completeness in Appendix G.6. It is worth noting that, despite its apparent simplicity, Proposition 2.7 has important implications for practical implementation—particularly concerning the direct use of `torch.nn.conv_transpose2d`. Moreover, it raises

considerations related to the preservation of orthogonality under conditions such as stride, groups, and dilation. For instance, transposed strided convolutions can be effectively utilized in the upsampling layers of U-Net architectures (Ronneberger et al., 2015). They can also be employed to design invertible neural networks for Normalizing Flows (see Appendix G.10).

**Native Grouped Orthogonal Convolutions:** Most modern CNNs use grouped convolutions (Howard, 2017; Xie et al., 2017; Liu et al., 2022). Beyond its advantages in terms of parameters and computational efficiency, it makes AOC more efficient as its parametrization can be parallelized, similarly as (Gorbunov et al., 2024; Liu et al., 2024). Given a group number $g$, the kernel of a grouped convolution $\mathbf{K} \in \mathbb{R}^{c_o \times \frac{c_i}{g} \times k \times k}$ can be viewed as a stack of $g$ kernels $\mathbf{K_i} \in \mathbb{R}^{\frac{c_o}{g} \times \frac{c_i}{g} \times k \times k}$, each constructed independently. Note that $c_o$ and $c_i$ must be multiple of $g$.

**Proposition 2.8** (Grouped Orthogonal Convolution). *A grouped convolution composed of $g$ kernels $(\mathbf{K_i})_g$ is orthogonal if and only if each individual convolution of kernel $\mathbf{K_i}$ is orthogonal.*

The proof of Proposition 2.8 is provided in Appendix G.7 and relies on the fact that the Toeplitz matrix of a grouped convolution is block diagonal. Note that when $g = c_i = c_o$, each kernel $\mathbf{K}_i$ has a single channel $c_i = c_o = 1$ that cannot be built with BCOP which requires $c \geq 2$.

**Native Orthogonal Convolutions with Dilation:** Introduced by (Yu & Koltun, 2016), Dilation is an effective means to increase the receptive field of a convolution without increasing its number of parameters or its computational cost. In (Su et al., 2022), the authors stated in an appendix that "any filter bank that is orthogonal for a standard convo-

lution is also orthogonal for a dilated convolution and vice versa." While this is mathematically accurate, it is essential to note that circular padding must be adjusted accordingly to remain within the scope of their theorem. Our method thus also supports orthogonal convolution with dilation. A description is given in Appendix G.8.

### 2.3. Efficient implementation of AOC

Beyond a mathematical framework that unlocks a more flexible use of orthogonal convolutions, we propose several design choices for an implementation that scales well to larger kernels, larger images and larger batch sizes. Although AOC includes the construction of BCOP and RKO kernels, our implementation improves the original ones at many stages. It results in an **8x** reduction of the original overhead in realistic settings. The implementation of AOC and several additional layers described in Appendix E is provided as an Open-source library called Orthogonium. Other implementation details and empirical testing are discussed in Appendices C and D.

**Fast implementation of the Block-convolution.** To the best of our knowledge, the only differentiable implementation of the BCOP method is available in the reference code (Li et al., 2019). However, since there is no cuda kernel available for Block-convolution, the authors relied on nested loops to perform all matrix multiplication to compute the resulting kernel. We propose an efficient and parallelized implementation with a single operation. Inspired by (Wang et al., 2020) and (Delattre et al., 2023), which aimed to compute $A \circledast A^T$ to prove orthogonality, we propose to replace the computation $B \circledast A$ by a convolution with zero padding between $B$ and $A^T$. This approach can also be seen as a specific case of convolutional einsum (Rabbani et al., 2024). This operation can be rewritten by re-ordering the summation to use a 2D convolution at its core. The strategy is to use the 2D convolution to compute one output filter. Then, the batch dimension can be used to compute all output filters in parallel. The code is detailed in the Fig. 2a.

**Reducing time complexity of BCOP.** Beyond the efficient parallelism of the $\circledast$ operation, we propose to parallelize the whole kernel computation: the parametrization can be seen as the composition of many small kernels. Not only can those be created in parallel, but they can also be combined efficiently. As the $\circledast$ is an associative operation (Proposition G.1), we can leverage the parallel associative scan (Zouzias & McColl, 2023; Hillis & Steele Jr, 1986) to parallelize the iterations of the original algorithm. The original $2*(k-s)$ sequential $\circledast$ operations can then be done in $\mathcal{O}(\log(k-s))$ iterations (Fig. 2b). This is unlocked in practice if $\circledast$ implementation supports batching. We propose to achieve this by using grouped `Conv2d` implementation: by concatenating $g$ kernels and setting groups $= g$, we can compute the batched $\circledast$ in parallel.

**Efficient implementation.** By examining Eq. (8), one observes that, depending on the values of $s$ and $k$, the parametrization can be simplified to either $\mathbf{K}_{AOC} = \mathbf{K}_{\text{BCOP}}$ or $\mathbf{K}_{AOC} = \mathbf{K}_{\text{RKO}}$. While BCOP is generally not suited for handling stride directly, we have identified specific cases – namely when $c_i < c_o$ – where stride can indeed be applied directly to a BCOP kernel (see Appendix G.9) without requiring the full parametrization. Although not proposed in (Li et al., 2019), this observation refines our overall characterization of BCOP's limitations with stride, showing that exceptions exist under certain conditions. The complete decision tree used in our implementation is shown in Fig. 2c, with each branch's orthogonality proved in Appendices G.4, G.5 and G.9.

## 3. Evaluation and applications

In this section, we evaluate the expressiveness and scalability of AOC convolutions. To assess expressiveness, we require a benchmark that also accounts for the orthogonality property. We evaluate this within the framework of certifiable robustness using 1-Lipschitz constrained networks, where enforcing orthogonality tightens Lipschitz estimation, improving robustness. AOC strictly represents 1-Lipschitz functions, crucial for Lipschitz-constrained networks. The overall Lipschitz constant of a sequence of layers is typically estimated as the product of the individual layer constants; however, this bound is often loose, and computing the exact constant is NP-hard (Virmaux & Scaman, 2018). (Anil et al., 2019) shows that combining MaxMin activation with orthogonal layers makes the aforementioned bound tight. (Bartlett et al., 2017) showed that if $f$ is a 1-Lipschitz classification function and $l$ is the label index, a lower bound on robustness radius $\epsilon$ in $L_2$ norm at $x$ is:

$$\epsilon \geq \frac{f(x)_l - \max_{i \neq l} f(x)_i}{\sqrt{2}}$$

This certificate is independent of adversarial attacks, ensuring robustness remains unchanged even if new attacks arise. It is also computationally cheap and optimizable in a loss function (Singla & Feizi, 2022; Hu et al., 2023).

To showcase AOC's expressiveness and scalability, we conduct experiments on CIFAR-10, a widely used benchmark for certifiable robustness, and ImageNet, a large-scale dataset. We train two networks per task: one prioritizing accuracy-noted *"AOC accurate"* in Table 2- and the other robustness-denoted *"AOC robust"*-enabling a thorough evaluation. Finally, the training recipe of (Hu et al., 2023) was used with original convolutions replaced by AOC, showcasing its expressiveness with extra data. This last recipe is denoted *"AOC robust*"*. For scalability, we compare AOC's computational overhead on a standard architecture, assessing its feasibility in large-scale applications.

| | Models | Accuracy | Provable Accuracy $\epsilon = \frac{36}{255}$ | Trainable Parameters |
|---|---|---|---|---|
| CIFAR-10 | BCOP | 72.2 | 58.2 | 2.6M |
| | GloRo[†] | 77.0 | 58.4 | 8.0M |
| | Local-Lip-B[†] | 77.4 | 60.7 | 2.3M |
| | Cayley Large | 74.6 | 61.4 | 21.0M |
| | SOC 20 | 78.0 | 62.7 | 27.0M |
| | SOC+ 20 | 76.3 | 62.6 | 27.0M |
| | CPL XL[†] | 78.5 | 64.4 | 236.0M |
| | AOL Large[†] | 71.6 | 64.0 | 136.0M |
| | SLL Small[†] | 71.2 | 62.6 | 41.0M |
| | SLL Medium[†] | 72.2 | 64.3 | 78.0M |
| | SLL Large[†] | 72.7 | 65.0 | 118.0M |
| | SLL X-Large[†] | 73.3 | 65.8 | 236.0M |
| | Li-Resnets[†*] | 82.1 | 70.1 | 49.0M |
| | Li-Resnets++[†*] | 87.0 | 78.1 | 49.0M |
| | **AOC** accurate | 91.5 | 00.0 | 2.3M |
| | **AOC** robust | 74.0 | 64.3 | 41.3M |
| | **AOC** robust* | 85.3 | 75.0 | 46.3M |
| IN-1K | Li-Resnets[†*] | 45.6 | 35.0 | 86.0M |
| | Li-Resnets++[†*] | 49.0 | 38.3 | 86.0M |
| | **AOC** accurate | 68.2 | 00.0 | 53.1M |
| | **AOC** robust | 42.1 | 26.3 | 53.1M |

Table 2: **AOC is competitive on small-scale datasets and enables affordable training on large-scale datasets.** For both cifar-10 (top) and Imagenet-1K (bottom), we reported provable accuracy (accuracy guaranteed under any attack whose $l_2$ radius is $\frac{36}{255}$) from respective papers. we evaluate our models under two settings: one tailored for accuracy and another for robustness. * denotes the use of extra data. Our networks are trained under two settings that use similar architectures but differ in their loss parameters (see Appendix H).[†] denotes nonorthogonal methods.

### 3.1. Certifiable robustness with 1-Lipschitz Networks.

Detailed architectures and training parameters are detailed in Appendix H. We report in Table 2 both the standard and provable accuracy[2] and compare with previous methods. We will highlight the interest of AOC through the 3 following observations.

**Observation 1: AOC allows construction of expressive 1-Lipschitz networks.** While AOC does not improve the expressiveness of its original building blocks (namely BCOP)[3], its flexibility unlocks the construction of complex blocks as depicted in Fig. 5a. Such a block permits the training of 1-Lipschitz networks that achieve competitive performance

with similar sizes and similar training cost as their unconstrained counterparts; The *"AOC accurate"* training configuration (in Table 2) achieves more than 90% clean accuracy on CIFAR-10 using less than 3M parameters, showing that AOC is expressive. This comes at the cost of 0% provable accuracy (certificates smaller than the radius $\epsilon = 36/255$): regardless of its resilience to empirical adversarial attacks. Conversely, the *AOC robust* provides a high certifiable accuracy (64.3%), but lower clean accuracy (74% instead of 91.5%), illustrating the well-known accuracy-robustness trade-off (Béthune et al., 2022). Finally, networks trained with AOC do not use normalization techniques such as batch normalization (Ioffe & Szegedy, 2015) or layer normalization (Ba et al., 2016).

**Observation 2: Old methods become competitive with scale.** Scaling the original BCOP network of (Li et al., 2019) from 2.6M to 41.3M parameters makes this method competitive with more recent approaches ($+6\%$ on provable accuracy). This is especially notable as our experiments did not use techniques such as last layer normalization(Singla & Feizi, 2022), certificate regularization (Singla & Feizi, 2021b), or DDPM augmentation (Hu et al., 2023). This was possible only due to the fast implementation of AOC, which enables larger batch sizes and faster training.

**Observation 3: State of the art aligns with computational budget.** Our experiments align with the work of (Prach & Lampert, 2024; Bubeck & Sellke, 2021), who noted that robust training does not have the same scaling laws as standard training: in order to obtain robustness, alongside accuracy and generalization, more data and larger architectures are required by an order of magnitude greater than was is currently being done. This explains why (Hu et al., 2023) used 4.5 million of images for their results on CIFAR-10, highlighting the need for scalable implementations that can handle this trend.

### 3.2. Scalability of AOC

As observed by (Prach et al., 2024), a slow implementation leads to increased training time and, consequently, lower performances in practical contexts. In this section, we demonstrate that AOC offers a key advantage: its computational cost does not depend on the input size or shape, making it well-suited for large-scale datasets, where handling large images is crucial. Although other methods may perform better on smaller datasets, they struggle to scale to widely used architectures like ResNet-34 (He et al., 2016), as shown in Table 3. On the other end, our method's low memory cost enables larger batch sizes, and since our parameterization is batch-size independent, the overhead decreases as batch size increases. Ultimately, this results in a training time only 13% slower than its unconstrained counterpart. The detailed protocol is detailed in Appendix H.1.

---

[2]proportion of samples with a certificate greater than $\epsilon$
[3]see Appendix I for the reproduction of their results

| Name | Batch Size | Train Time (ms) | Train Memory (GB) | Test Time (ms) | Test Memory (GB) |
|---|---|---|---|---|---|
| Conv2D (ref) | 128 | 137 (1.00x) | 4.7 (1.00x) | 50 (1.00x) | 1.4 (1.00x) |
| AOC (ours) | 128 | **239 (1.75x)** | **5.3 (1.15x)** | **53 (1.06x)** | **1.4 (1.02x)** |
| BCOP | 128 | 389 (2.85x) | 8.6 (1.84x) | 62 (1.25x) | 1.5 (1.06x) |
| SOC | 128 | 664 (4.86x) | 12.8 (2.73x) | 429 (8.55x) | 1.8 (1.30x) |
| Cayley | 128 | 584 (4.27x) | 19.0 (4.07x) | 247 (4.94x) | 2.0 (1.45x) |
| Conv2D (ref) | 256 | 284 (1.00x) | 9.1 (1.00x) | 91 (1.00x) | 2.7 (1.00x) |
| AOC (ours) | 256 | **354 (1.25x)** | **9.8 (1.07x)** | **97 (1.06x)** | **2.7 (1.01x)** |
| BCOP | 256 | 624 (2.20x) | 13.9 (1.53x) | 135 (1.48x) | 2.8 (1.03x) |
| Conv2D (ref) | 512 | 550 (1.00x) | 17.9 (1.00x) | 172 (1.00x) | 5.3 (1.00x) |
| AOC (ours) | 512 | **622 (1.13x)** | **18.6 (1.04x)** | **176 (1.02x)** | **5.4 (1.01x)** |
| BCOP | 512 | 1116 (2.03x) | 24.6 (1.38x) | 256 (1.48x) | 5.4 (1.02x) |

Table 3: **AOC benefits from scale.** As demonstrated on a ResNet-34, previous methods impose significant overhead when input images are large ($224 \times 224$). In contrast, since our method's computational cost is independent of layer input size, its overhead decreases as batch size increases. Furthermore, the low memory overhead enables larger batches at scale.

## 4. Conclusion and Broader Impact

In this paper, we introduced AOC, a method that combines two existing approaches in order to build orthogonal convolutions that support essential features such as stride, transposition, groups, and dilation. Our results demonstrate that this layer is both expressive and scalable. Beyond its standalone benefits, our framework enhances existing layers: In Appendix E, we integrate our method with SLL (Araujo et al., 2023) to build an efficient downsampling residual block, propose optimizations to reduce the memory footprint of SOC (Singla & Feizi, 2021a), and present a strategy to make Sandwich layers (Wang & Manchester, 2023) scalable for convolutions. Besides, AOC, by unlocking features such as transposition, grouping, or dilation, paves the way for future work leveraging orthogonality in varied architectures and applications, such as upsampling convolutions for U-Nets (Ronneberger et al., 2015), VAEs (Kingma, 2013), WGANs and Optimal transport (Arjovsky et al., 2017; Béthune, 2024), or reduced complexity (Li et al., 2018; Xie et al., 2017). In support of further research and development, we open sourced our implementation in the Orthogonium library.

## Impact Statement

This paper presents work whose goal is to advance the field of Machine Learning. There are many potential societal consequences of our work, none which we feel must be specifically highlighted here.

## Acknowledgement

This work was carried out within the , DEEL project which is part of IRT Saint Exupéry and the ANITI AI cluster. The authors acknowledge the financial support from DEEL's Industrial and Academic Members and the France 2030 program – Grant agreements n°ANR-10-AIRT-01 and n°ANR-23-IACL-0002.

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

## Contents

## A. Applications of orthogonal convolutions

Orthogonal layers have become fundamental components in various deep learning architectures due to their unique mathematical properties, which benefit multiple applications.

**Provable Robustness with 1-Lipschitz Networks.**    Ensuring robustness against adversarial attacks is a critical challenge. Early work in this field (Szegedy et al., 2014) identified a link between a network's adversarial robustness and its Lipschitz constant, which led to the development of networks with a Lipschitz constant of one (1-Lipschitz networks). Initially, regularization techniques were used (Cisse et al., 2017), but interest in constrained networks quickly grew. Orthogonal layers, in particular, have an inherent Lipschitz constant of one, as they preserve the input norm through each transformation. This property is instrumental in achieving provable robustness by allowing for certified bounds on the network's output perturbations in response to adversarial inputs (Anil et al., 2019). By controlling the network's sensitivity to input changes, orthogonal layers play a crucial role in building models resilient to adversarial manipulations. Besides robustness, Lipschitz-constrained networks have diverse applications: they are closely linked to WGANs and optimal transport (Arjovsky et al., 2017; Béthune, 2024), enable scalable differential privacy (Béthune et al., 2024), allow conformal prediction in adversarial setting (Massena et al., 2025), and prevent singularities in diffusion models (Yang et al., 2023). Additionally, they guarantee the existence and uniqueness of solutions in classification (Béthune et al., 2022) and flow-matching networks (Perko, 2013; Coddington et al., 1956).

**Enhancing Performance in Normalizing Flows.**    Normalizing flows are a class of generative models that transform simple probability distributions into complex ones through a series of invertible and differentiable mappings. Although they have different objectives, this domain intersects with the field of provable adversarial robustness. For instance, (Dinh et al., 2017; Rezende & Mohamed, 2015) employs a channel masking scheme, which was later used to emulate striding in Lipschitz layers designed for adversarial robustness. Separately, Lipschitz layers can be applied to build invertible residual networks (Behrmann et al., 2019). In both fields, orthogonal convolutions are essential, as they facilitate the construction of invertible transformations with tractable Jacobian determinants. The use of orthogonal layers ensures that the Jacobian determinant is constant or easily computable, simplifying likelihood estimation during training (Kingma & Dhariwal, 2018). This property enables efficient and stable training of normalizing flow models, leading to improved performance in density estimation and generative tasks.

**Stabilizing Training in Deep and Recurrent Neural Networks.**    Training recurrent neural networks (RNNs) involves propagating gradients through time, which can lead to vanishing or exploding gradients due to the multiplicative nature of sequential weight applications. Orthogonal weight matrices in RNNs help preserve the gradient norm across time steps, thus preventing degradation of the learning signal (Kiani et al., 2022). By constraining recurrent weights to be orthogonal,

the network maintains a consistent flow of information, enabling it to capture long-term dependencies more effectively. This stabilization is essential for tasks requiring understanding long-range dependencies in time series, such as language modeling and speech recognition. These approaches also facilitate the training of very deep networks (Qi et al., 2020) with improved generalization properties (Bansal et al., 2018).

**Improving Stability in Wasserstein Generative Adversarial Networks.** Generative Adversarial Networks (GANs) are powerful models for generating realistic data, but they often suffer from training instability. Wasserstein GANs (WGANs) (Arjovsky et al., 2017) address this issue by optimizing the Wasserstein distance between the real and generated data distributions. A key requirement for WGANs is that the discriminator (or critic) function must be Lipschitz continuous. Orthogonal layers naturally satisfy this Lipschitz condition, eliminating the need for techniques like weight clipping or gradient penalties (Gulrajani et al., 2017), which can adversely affect training dynamics. By incorporating orthogonal convolutions into the discriminator, WGANs achieve more stable training (Miyato et al., 2018; Müller et al., 2019) and produce higher-quality generative results. Orthogonality has been integral to the successful scaling of GAN training (Brock et al., 2018).

**Extending the applicability of Proximal Neural Networks.** Proximal Neural Networks (PNNs) offer a principled approach to designing and training deep architectures by drawing inspiration from optimization theory—specifically, proximal algorithms and $\alpha$-averaged operators (Hasannasab et al., 2020; Combettes & Pesquet, 2020). A central insight in this framework is the emergence and enforcement of orthogonality in learned operators, which can improve network stability. (Hertrich et al., 2021) proposes an extension to convolutional operators but notes that, for limited kernel sizes, optimizing on the manifold is challenging and instead relies on penalization, similar to (Wang et al., 2020). Using our method, AOC, and its transpose could overcome this limitation.

# B. BCOP and SC-Fac re-explained with our framework

In this section, we outline the key steps in constructing orthogonal convolutions using the BCOP (Fig. 1) or SC-Fac methods within the framework established in the paper.

**From Orthogonal Matrices to Orthogonal $c_o \times c_i \times 1 \times 1$ Convolutions.** A substantial body of research exists on building orthogonal matrices $M \in \mathbb{R}^{c_o \times c_i}$. One common approach involves applying a differentiable projection operator to an unconstrained weight matrix, yielding an orthogonal matrix such that $MM^T = I$ or $M^T M = I$. Various methods exist, including the Björck and Bowie orthogonalization scheme (Björck & Bowie, 1971), the exponential method (Singla & Feizi, 2021b), the Cayley method (Trockman & Kolter, 2021), and QR factorization (Van Den Berg et al., 2018). An orthogonal matrix can easily be reshaped into a convolution kernel with a $1 \times 1$ kernel $\mathbf{M} \in \mathbb{R}^{c_o \times c_i \times 1 \times 1}$, and such a convolution is orthogonal if $M$ is orthogonal. These convolutions are mainly used to change the number of channels (Fig. 1-1).

**Building $c \times c \times 1 \times 2$ Orthogonal Convolutions.** Stacking two orthogonal $1 \times 1$ convolution kernels along their last dimensions[4] leads to a $1 \times 2$ convolution, though it is generally not orthogonal. Authors of (Xiao et al., 2018; Su et al., 2022) noted that additional constraints are needed, proposing a half-rank symmetric projector to construct a $1 \times 2$ orthogonal convolution: from a column-orthogonal matrix $M \in \mathbb{R}^{c \times \frac{c}{2}}$[5], the matrix $N = MM^T \in \mathbb{R}^{c \times c}$ is a symmetric projector that satisfies:

$$N = N^2 = N^T \quad \text{and} \quad (I - N) = (I - N)^2 = (I - N)^T$$

These two matrices can be reshaped into $c \times c \times 1 \times 1$ convolution kernels, and stacking them along the last dimension results in an orthogonal $c \times c \times 1 \times 2$ convolution kernel:

$$\mathbf{P} = \texttt{stack}([\mathbf{N}, \mathbf{I} - \mathbf{N}], \texttt{axis} = -1) \Rightarrow \mathbf{P} \circledast \mathbf{P}^T = \mathbf{I}$$

Similarly, stacking along the penultimate dimension, $\mathbf{Q} = \texttt{stack}([\mathtt{N}, \mathtt{I} - \mathtt{N}], \texttt{axis} = -2)$, results in an orthogonal $c \times c \times 2 \times 1$ kernel. Although already proven by previous work, proof of this can be found in Appendix G.3.

**From $1 \times 2$ to $k_1 \times k_2$ Orthogonal Convolutions.** The three papers propose to build standard orthogonal convolutions by composing smaller kernels based on the following properties:

---

[4]Done in practice with `torch.stack([K_1, K_2], axis=-1)`
[5]This implies that $c \geq 2$. When $c$ is even, $\lfloor \frac{c}{2} \rfloor$ is used in practice

Using Block-convolution, we can represent the composition of $1 \times 2$ and $2 \times 1$ kernels[6] to obtain a kernel with any desired shape. The differences among the three approaches lie in the composition order: Authors of (Su et al., 2022) chose to compose $(k_1 - 1)$ $2 \times 1$ kernels $\mathbf{P_i}$, followed by a $1 \times 1$ kernel $\mathbf{M}$, and $(k_2 - 1)$ $1 \times 2$ kernels $\mathbf{Q_i}$ to form a $k_1 \times k_2$ kernel:

$$\mathbf{K}_{\text{SC-Fac}} = \underbrace{\mathbf{P_{k_1-1}} \circledast \ldots \circledast \mathbf{P_1}}_{\text{all 1x2 kernels}} \circledast \underbrace{\mathbf{M}}_{\text{1x1}} \circledast \underbrace{\mathbf{Q_1} \circledast \ldots \circledast \mathbf{Q_{k_2-1}}}_{\text{all 2x1 kernels}}$$

On the other hand, authors of (Li et al., 2019)(Xiao et al., 2018) alternated $2 \times 1$ and $1 \times 2$ kernels, ending with a $1 \times 1$ convolution:

$$\mathbf{K}_{\text{BCOP}} = \underbrace{(\mathbf{P_{k-1}} \circledast \mathbf{Q_{k-1}})}_{\text{pairs of 1x2 and 2x1 kernels}} \circledast \ldots \circledast (\mathbf{P_1} \circledast \mathbf{Q_1}) \circledast \underbrace{\mathbf{M}}_{\text{1x1}}$$

Both approaches have incomplete parametrizations: the first is limited to separable convolutions, while the second shows counterexamples in the general 2D convolution case. However, both methods use the same number of parameters for a given kernel size. Building a complete parametrization of 2D convolutions remains an open question, discussed in Appendix F. We thus base our work on the BCOP parametrization (Li et al., 2019) for two main reasons: (1) any $2 \times 2$ convolution not parametrizable by BCOP can be represented with a $3 \times 3$ kernel – a feasible solution given the trend toward larger kernels (Trockman & Kolter, 2024; Ding et al., 2022); (2) BCOP enables a faster and less memory-intensive implementation (see Section 2.3), unlocking larger networks that compensate for any potential expressiveness loss.

## C. Impact of the orthogonalization scheme on AOC

In this section we will discuss on of the design choices for AOC: the orthogonalization procedure. Many approaches exists and we will describe the main approaches:

### C.1. Non exhaustive list of orthogonalization methods

#### C.1.1. QR FACTORIZATION VIA MODIFIED GRAM-SCHMIDT

The QR factorization is obtained using the Modified Gram-Schmidt (MGS) algorithm, as initially proposed in (LaPlace, 1820). The MGS procedure follows the same fundamental computational steps as the classical Gram-Schmidt method; however, it executes these steps in a different order. In classical Gram-Schmidt, each iteration involves computing a sum that includes all previously computed vectors. Conversely, the modified version corrects numerical errors at each step, leading to improved numerical stability.

---

**Algorithm 1** QR Factorization via Modified Gram-Schmidt

---

**Require:** $W = [w_i]_{i \in [0, C-1]} \in \mathbb{R}^{C \times C}$
**Ensure:** $W = QR$, where $Q \in \mathbb{O}(C)$ is an orthogonal matrix and $R \in \mathbb{R}^{C \times C}$ is an upper triangular matrix.
  1: **procedure** MODIFIED GRAM-SCHMIDT($W$)
  2:     $w_i^c = w_i, \forall i \in [0, C-1]$                                                $\triangleright$ Initialize with a copy of $W$
  3:     **for** $j \in range(C)$ **do**
  4:         $r_{j,j} = \|w_j^c\|_2$
  5:         $q_j = w_j^c / r_{j,j}$
  6:         **for** $k \in range(j+1, C)$ **do**
  7:             $r_{j,k} = q_j^T w_k^c$
  8:             $w_k^c = w_k^c - r_{j,k} q_j$
  9:         **end for**
10:     **end for**
11:     **return** $QR$
12: **end procedure**

---

---

[6]As indicated in Definition 2.2, $k_1$ and $k_2$ parameters do not affect row or column orthogonality

### C.1.2. CAYLEY TRANSFORM

The Cayley transform (Cayley, 1846) provides an alternative approach for constructing orthogonal matrices. It exploits the bijective relationship between the special orthogonal matrices $Q$ (i.e., orthogonal matrices with determinant $+1$) and skew-symmetric matrices $A$ (satisfying $A^T = -A$). The transformation is defined as follows:

$$Q = (I - A)(I + A)^{-1} \quad \text{or} \quad Q = (I + A)^{-1}(I - A). \tag{12}$$

A matrix $A$ can be naively constructed as a skew-symmetric matrix using:

$$A = M^T - M. \tag{13}$$

The extension of the Cayley transform to rectangular matrices is detailed in (Pauli et al., 2023) and is formalized in Algorithm 2.

---

**Algorithm 2** Cayley Transform Algorithm

---

**Require:** $W \in \mathbb{R}^{M \times C}$, where $M > C$
**Ensure:** $\hat{W} \in \mathbb{R}^{M \times C}$, an orthogonal matrix
1: **procedure** CAYLEY TRANSFORM($W$)
2:     $U, V = W[:C, :], W[C:, :]$               ▷ Partition $W$ such that $U \in \mathbb{R}^{C \times C}$ and $V \in \mathbb{R}^{(M-C) \times C}$
3:     $A = U - U^T + V^T V$                                        ▷ Note: $A$ is not strictly skew-symmetric
4:     $B = (I + A)^{-1}$
5:     $\hat{W}_1 = B(I - A)$                                                                   ▷ $\hat{W}_1 \in \mathbb{R}^{C \times C}$
6:     $\hat{W}_2 = -2VB$                                                                  ▷ $\hat{W}_2 \in \mathbb{R}^{(M-C) \times C}$
7:     $\hat{W} = [\hat{W}_1, \hat{W}_2]$                                        ▷ Concatenation yields $\hat{W} \in \mathbb{R}^{M \times C}$
8:     **return** $\hat{W}$
9: **end procedure**

---

This approach involves matrix inversion, which can be computationally expensive, particularly for large matrices.

### C.1.3. EXPONENTIAL MAP

The exponential map (Singla & Feizi, 2021b) is another technique used to generate orthogonal matrices, leveraging the properties of skew-symmetric matrices. The matrix exponential of $A$ is defined as:

$$\exp(A) = \sum_{k=0}^{\infty} \frac{A^k}{k!}. \tag{14}$$

For a skew-symmetric matrix $A$, it can be shown that:

$$(e^A)^T = e^{-A}. \tag{15}$$

Since the product of a matrix and its transpose is the identity matrix, this ensures that $e^A$ is orthogonal.

In practice, numerical computations only approximate the exponential function by summing a finite number of terms. Algorithm 3 describes the Exponential Map with a spectral normalization to prevent floating-point overflows.

This method parametrizes only the special orthogonal group. The proof follows from the determinant properties of matrix exponentiation.

### C.1.4. CHOLESKY DECOMPOSITION

The Cholesky decomposition has also been employed for orthogonalization (Hu et al., 2023). Given a weight matrix $M$, we construct the covariance matrix:

$$C = MM^T. \tag{16}$$

---

**Algorithm 3** Exponential Map Algorithm

---

**Require:** $W \in \mathbb{R}^{C \times C}$, $p \in \mathbb{N}$ (number of terms in the expansion)
**Ensure:** $\hat{W} \in \mathbb{R}^{C \times C}$, an orthogonal matrix
 1: **procedure** LIPSCHITZ EXPONENTIAL($W$)
 2:      $A = W - W^T$
 3:      $\hat{A} = A/\|A\|_2$                                                         ▷ Spectral normalization
 4:      $\hat{W} = I_C, \hat{A}_k = I_C$
 5:      **for** $j \in range(1, p)$ **do**
 6:          $\hat{A}_k = \frac{\hat{A}_k}{k} \times \hat{A}$
 7:          $\hat{W} = \hat{W} + \hat{A}_k$
 8:      **end for**
 9:      **return** $\hat{W}$
10: **end procedure**

---

Since $C$ is positive semi-definite by construction, its Cholesky decomposition exists:

$$C = LL^T. \tag{17}$$

Solving the triangular system $LW = M$ yields an orthogonal matrix $W$. To ensure the positive-definiteness of $C$, a small positive perturbation is added before decomposition (Hu et al., 2023).

This method is computationally efficient, provided that numerical stability is carefully managed.

### C.1.5. BJÖRCK AND BOWIE

(Björck & Bowie, 1971) proposed an iterative algorithm for computing the best orthogonal approximation of a given matrix. This process relies solely on matrix multiplications:

$$W_{t+1} = (1 + \beta)W_t + \beta W_t W_t^\top W_t.$$

This algorithm corresponds to the gradient descent of the regularization term $\|WW^\top - I\|_2$ with a learning rate of $\beta$. The convergence radius is 1, determined by the spectral norm of $W$. To ensure numerical stability, our implementation applies spectral normalization to the matrix. Under these conditions, $\beta$ can be increased to its theoretical maximum of $\frac{1}{2}$, which accelerates convergence.

This specific setting allows for rapid convergence of the algorithm, requiring only 25 iterations, as demonstrated in (Anil et al., 2019). Additionally, our unit testing scheme reveals that the number of iterations can be reduced to 12 without significant loss of orthogonality. This is achieved through our efficient implementation of the power iteration method, which retains an estimate of the dominant eigenvector across iterations.

### C.2. Impact on AOC

Since AOC extensively uses orthogonalization algorithms, it is reasonable to question the original choice made by (Li et al., 2019; Serrurier et al., 2024) to use the Björck and Bowie algorithm. Our implementation allows the use of multiple algorithms, with the currently implemented: exponential method, QR scheme, Björck and Bowie, and Cholesky method. We trained the same network as in (Li et al., 2019) with changing only the orthogonalization method, results are shown in Fig. 3. We observed 3 distinct groups of methods:

- exponential: Although one of the most efficient methods, AOC use mainly rectangular matrices. This case was handled using padding, which made the method much slower and hindered optimization.

- Björck and Bowie & QR: these methods perform similarly both in terms of speed and accuracy.

- cholesky: is significantly faster in this setting but leads to a lower accuracy. The right graph shows that, in fine, advantages and drawbacks of this approach cancel each other out. However, our unit testing scheme has shown that

AOC does not pass the unit tests described in Appendix D with expected tolerance, needing a relaxed tolerance of $5 \times 10^{-2}$.

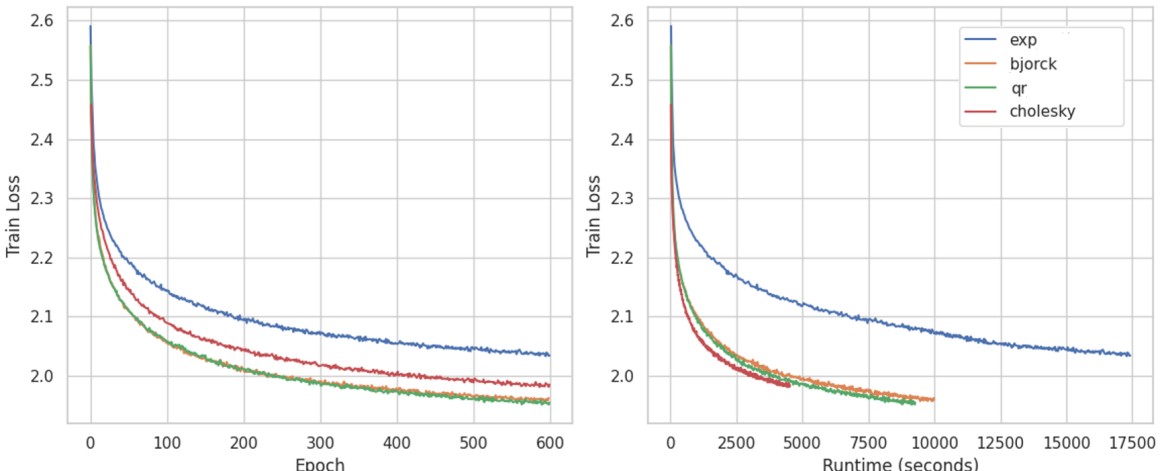

Figure 3: Impact of the orthogonalization method on AOC: in the same setup as (Li et al., 2019) we observe that the orthogonalization precedure can have a significant impact on the training time and training accuracy. Except from exponential method which suffers from limitations due to our way to handle non-square matrices, other methods shows similar performance given the same computation time.

Given these observations, we sticked to the original choice of Björck and Bowie algorithm, as it is GPU and mixed precision compliant. Thanks to our unit-testing scheme, we observed that once the matrix is constrained to be 1-Lipschitz, we can set $\beta = 0.5$ (the maximum value for which convergence can be guaranteed). This allows us to lower the number of iterations to 12 without affecting orthogonality.

## D. Empirical evaluation of the Lipschitz constant of our method

**Evaluating the Lipschitz constant of a network**  Beyond the creation of a constrained layer, the evaluation of the Lipschitz constant of a layer is by itself an active field: early work used fast Fourier transform to evaluate a lower bound of the Lipschitz constant of a convolutional layer with circular padding (Sedghi et al., 2018). This work was later improved with a method that is quicker (Senderovich et al., 2022), supports other types of padding (Grishina et al., 2024), or allows the extraction of a larger part of the spectrum (Boroojeny et al., 2024). The work of (Delattre et al., 2023) (Delattre et al., 2024) allows us to compute a certifiable upper bound efficiently under different types of padding. It is worth recalling that inferring the global Lipschitz constant of a network given the Lipschitz constant of each layer is an NP-Hard problem(Virmaux & Scaman, 2018). Then, (Pauli et al., 2024; Fazlyab et al., 2019; Wang et al., 2024) aim to tackle using SDP (Semi-definite programming) tools. Our work can also contribute to this issue as the orthogonal layer allows a tighter product bound (ie. bound using the product of the Lipschitz constant of each layer to evaluate the constant of the whole network).

**The need for an empirical evaluation of the Lipschitz constant of AOC.**  Despite the theoretical guarantees ensuring orthogonality in our construction, empirical checks are necessary to confirm implementation correctness. Such verification prevents two types of issues:

1. **Checking of numerical instabilities:** Issues arising from floating-point precision, such as those introduced by small epsilon values added to avoid division by zero.

2. **Checking for implementation discrepancies:** Differences between mathematical formalism and its translation to popular frameworks (e.g., SOC proofs assume circular padding, while its implementation uses zero padding).

**Checking the orthogonality of a layer under stride, group, transposition, and dilation conditions.**  The numerical stability and the convergence of an orthogonal layer is dependent on the training hyper-parameters: mainly the number of

iterations used in most methods, but the learning rate and weight decay can also play a significant role. We then need an evaluation method that scales along with the convolution and that can be used at the end of each training. On the other hand, as scalable methods can be imperfect, we also need a method that computes very precise bounds without making any assumptions on the layer parameters (like padding, or stride). In order to overcome this, we tested our layers with two distinct methods:

1. **Explicit SVD on Toeplitz Matrices:** Using the impulse response approach, we construct the Toeplitz matrix for any padding and stride, allowing direct computation of singular values. This method, though accurate, is computationally expensive for large input images.

2. **Product Bound for BCOP and RKO Kernels:** The upper bound for the BCOP kernel is computed using standard methods, while the SVD of the reshaped RKO kernel is used for direct evaluation.

**Unit testing of the implementation.** We used both of these two approaches in our unit tests. This enables us to ensure that the second method (which is faster and more scalable) is correct to check that our layer is effectively orthogonal. Also, our layer unlocks the use of the transposed convolution, which can be used to compute directly the equation of orthogonal layers:

$$(\mathcal{S}_s\mathcal{K})(\mathcal{S}_s\mathcal{K})^T = \mathcal{I} \quad \text{(row orthogonal)}$$
$$\text{conv}_{\mathbf{K}}(\text{ConvTranspose}_{\mathbf{K}}(x, \text{stride} = s), \text{stride} = s) = x$$

Naturally, the other direction can also be verified for column orthogonal layers.

To follow the optimization depicted in Fig. 2c, we tested each branch independently. For each branch, we tested multiple values for kernel size, stride, dilation, input channels, and output channels. For the kernel size, along with standard configurations of $3 \times 3$ and $5 \times 5$ kernels, we also covered cases for $1 \times 1$ kernels and even-sized kernels. For input/output channels, we covered various values and all the inequalities discussed in this paper (for instance when $c_o > c_i s^2$). We ran similar tests for transposed convolution. As the computation of the singular values using the explicit construction of the Toeplitz matrix is quite expensive, we used it on small $8 \times 8$ images, this is also a good way to check for padding issues, as the kernel size is not negligible with respect to the image size. All the checks over the singular values for both methods were done with a tolerance of $1e^{-4}$.

Finally, we tested independently the properties of the Block-convolution and the batched Block-convolution.

This amounts to 1442 tests that have the following repartition:

- block convolution: 640 tests

- convolution: 418 tests

  - common configurations in CNN: 72 tests
  - extended strided configurations: 150 tests
  - even kernel size: 24 tests
  - depthwise: 24 tests
  - kernel size = stride : 100 tests

- conv transpose: 384 tests

- RKO: 370 tests

This test bank was of precious use to confirm that all parameters can be combined together in practice. The total coverage of the library is 93%.

## E. Using the content of this paper to improve SLL, SOC, and Sandwich

In this section, we will explore how the content of this paper can be used to improve existing layers from the state of the art.

### E.1. Improving skew orthogonal convolution (SOC)

This method, introduced by (Singla & Feizi, 2022) uses the fact that an exponential of a skew-symmetric matrix is orthogonal. The initial implementation builds a skew-symmetric kernel and computes the exponential convolution. However, without proper tools to compute the exponential of a convolution kernel, this exponential was computed implicitly for each input by using the Taylor expansion of the exponential (see Eq. (18)).

**Theorem E.1** (Explicit conv exponential). *We can use Eq.* (4) *to compute explicitly the exponential of a kernel* **K**:

$$x + \frac{\mathbf{K} * x}{1!} + \frac{\mathbf{K} * \mathbf{K} * x}{2!} + \dots \tag{18}$$

$$= \left( Id + \mathbf{K} + \frac{\mathbf{K} \circledast \mathbf{K}}{2!} + \frac{\mathbf{K} \circledast \mathbf{K} \circledast \mathbf{K}}{3!} + \dots \right) * x \tag{19}$$

Equation (19) shows that we can compute the exponential of a convolution kernel a single time, while the formulation in Eq. (18) needs to be done for each input $x$. In other words, we can apply one conv instead of $n_{iter}$ convs. Note that the resulting kernel is then larger than the original one (as stated in Table 1). In theory, this could unlock large speedups, but the gain is limited in practice as the implementation of convolution layers is optimized for small kernels and large images (Ding et al., 2022). However, the original implementation requires the storage of $n_{iter}$ maps, whereas our implementation only one. This, in practice, unlocks larger networks and batch sizes.

Also, it is possible to handle a change in the number of channels and striding using a similar approach as AOC layers.

### E.2. Improving SDP-based Lipschitz Layers (SLL)

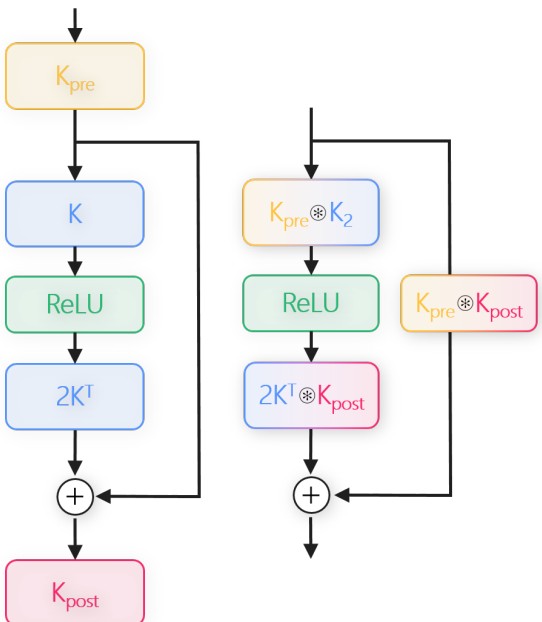

Figure 4: **The ⊛ can be used to enable $s \neq 1$ and $c_i \neq c_o$ configurations on SLL.** The flexibility of the ⊛ allows for operations resulting in a block with a similar structure as the original ResNet block.

SLL layer for convolutions, proposed in (Araujo et al., 2023), is a 1-Lipschitz layer defined as:

$$y = x - 2\mathbf{K}^T \star (\sigma(\mathbf{K} \star x + b))$$

Note that in the original paper, the equation is noted with product of two matrices $WT^{-\frac{1}{2}}$, for convolutions it represents toeplitz matrix, i.e. $WT^{-\frac{1}{2}} = \mathcal{K}$.

SLL layer does not natively support neither strides nor changes in the channel size. We propose to use the $\circledast$ to derive a block, based on SLL, that supports stride and $c_i \neq c_o$, and can replace the strided convolutions of the residual branch in architectures like ResNet.

A natural first step is to append a strided convolution after a SLL block. This layer, $conv_{K_{post}} \circ SLL$, can then be fused in the SLL block thanks to Proposition G.2:

$$\begin{aligned} y =& \mathbf{K}_{post} \star_s \left( x - 2\mathbf{K}^T \star \left( \sigma(\mathbf{K} \star x + b) \right) \right) \\ =& \mathbf{K}_{post} \star_s x - 2(\mathbf{K}_{post} \circledast \mathbf{K}^T) \star_s \left( \sigma(\mathbf{K} \star x + b) \right) \end{aligned}$$

This allows to build a block based on SLL and that supports stride and channel changes. However, this creates an asymmetry between the convolution before the activation and the one after the activation (that has a larger kernel size).

We propose also to add a second convolution before the SLL block, $conv_{K_{post}} \circ SLL \circ conv_{K_{pre}}$ allowing better control over the kernel size of each convolution:

$$\begin{aligned} y =& \mathbf{K}_{post} \star_s \mathbf{K}_{pre} \star x \\ &- 2(\mathbf{K}_{post} \circledast \mathbf{K}^T) \star_s \left( \sigma(\mathbf{K} \star \mathbf{K}_{pre} \star x + b) \right) \\ =& (\mathbf{K}_{post} \circledast \mathbf{K}_{pre}) \star_s x \\ &- 2(\mathbf{K}_{post} \circledast \mathbf{K}^T) \star_s \left( \sigma((\mathbf{K} \circledast \mathbf{K}_{pre}) \star x + b) \right) \end{aligned}$$

The proposed block is still a 1-Lipschitz layer (as a composition of 1-Lipschitz and orthogonal layers), and support efficiently strides and changes of kernel sizes. A visual description is provided in Fig. 4. This approach is more efficient than the explicit construction that uses 3 distinct convolutions, as kernels are merged once per batch, and intermediate activations of extra convolutions do not need to be stored backward. Typically, when $\mathbf{K}$, $\mathbf{K}_{pre}$ and $\mathbf{K}_{post}$ are $2 \times 2$ convolutions, this results in a residual block with two $3 \times 3$ convolutions in one branch and a single $4 \times 4$ convolution (with stride 2) in the second. This is very similar to transition blocks found in typical residual networks.

### E.3. Improving Sandwich Layers

Introduced by (Wang & Manchester, 2023), this approach aims to construct a 1-Lipschitz network globally rather than constraining each layer independently. In practice, this can be done either by (i) adding constraints between layers or (ii) creating layers that incorporate a non-linearity internally (a.k.a. sandwich layers). However, sandwich layers require an orthogonal matrix at their core. For convolutional layers, this is achieved by performing the orthogonalization of the layer in the Fourier domain, as described in the method from (Trockman & Kolter, 2021) and shown in their Algorithm 1.

---

**Algorithm 4** Sandwich convolutional layer (from (Wang & Manchester, 2023))

---

**Require:** $h_{\text{in}} \in \mathbb{R}^{p \times s \times s}$, $P \in \mathbb{R}^{(p+q) \times q \times s \times s}$, $d \in \mathbb{R}^q$
 1: $\hat{h}_{\text{in}} \leftarrow \text{FFT}(h_{\text{in}})$
 2: $\Psi \leftarrow \text{diag}(e^d)$, $\begin{bmatrix} \tilde{A} & \tilde{B} \end{bmatrix}^* \leftarrow \text{Cayley}(\text{FFT}(P))$
 3: $\hat{h}[:, i, j] \leftarrow \sqrt{2}\tilde{B}[:, :, i, j]\hat{h}_{\text{in}}[:, i, j]$
 4: $\hat{h} \leftarrow \text{FFT}(\sigma(\text{FFT}^{-1}(\hat{h}) + b))$
 5: $\hat{h}_{\text{out}}[:, i, j] \leftarrow \sqrt{2}\tilde{A}[:, :, i, j]\Psi\hat{h}[:, i, j]$
 6: $h_{\text{out}} \leftarrow \text{FFT}^{-1}(\hat{h}_{\text{out}})$

---

We can leverage AOC to construct the kernel of an orthogonal convolution, replacing the expensive operation performed in the Fourier domain. Thus, we can construct two kernels, $\mathbf{A}$ and $\mathbf{B}$, with appropriate constraints between the two and apply the rescaling and non-linearity directly in pixel space:

$$h_{\text{out}} = \sqrt{2}\mathbf{A}^\top \star \Psi\sigma\left(\sqrt{2}\Psi^{-1}\mathbf{B} \star h_{\text{in}} + b\right)$$

The use of the Fourier transform is costly for two reasons: first, it necessitates computation with complex values; and second, the cost of the operation depends on the input size, which can be prohibitive in large-scale settings with $224 \times 224$ images. Consequently, our approach can make such a layer more scalable.

### E.4. Extending Applicability to other methods.

Beyond the previously discussed approaches that show meaningful opportunities for improvement, our method can enhance a wide range of orthogonal convolutional layers. Specifically, we can incorporate our framework into any alternative orthogonal layers, enabling native support for strides in those layers. Furthermore, our approach can unlock features such as grouped convolutions, transposed convolutions, and dilations, broadening its utility and adaptability.

## F. About the incomplete parametrization of BCOP and SC-fac

As mentioned earlier, both BCOP and SC-Fac exhibit an incomplete parametrization. BCOP has an incomplete parametrization for 2D convolutions, while SC-Fac offers a complete parametrization but only for separable convolutions.

### F.1. Understanding the Limitations of BCOP

The limitations of BCOP parametrization have significant implications for its use in practical applications. Below, we provide a detailed discussion of the known limitations:

- The authors presented a counterexample involving a $2 \times 2$ convolution that is orthogonal but cannot be parametrized by BCOP. This highlights the incomplete nature of BCOP for parametrizing certain convolutional layers.

- However, this counterexample can be parametrized by a $3 \times 3$ BCOP convolution, which suggests that increasing the kernel size can potentially address the issue of incomplete parametrization.

- This does not imply that all $2 \times 2$ orthogonal convolutions can be parametrized using $3 \times 3$ BCOP convolutions, but it provides a useful starting point. It indicates that while BCOP may struggle with certain cases at smaller kernel sizes, increasing the kernel size could offer a pathway to improve coverage.

This problem is more complex than initially expected: the parametrization space of BCOP is disconnected. In simple terms, the disconnected nature of the parametrization space means that certain transformations cannot be continuously reached from others within the same parametrization framework. Nevertheless, a BCOP convolution with $c$ channels can have a connected component that represents all convolutions with $c/2$ channels. This property indicates that, although BCOP may not cover the entire space of orthogonal convolutions, it has subsets that can be effectively utilized for lower-dimensional problems.

Another noteworthy point is that the disconnected nature of the parametrization space could limit the efficiency of optimization algorithms that rely on continuous transformations during training. In practice, this means that certain optimal configurations may not be reachable through gradient-based methods, which could hinder the overall performance of models employing BCOP convolutions.

The issue of incomplete parametrization can be mitigated by increasing the number of channels and kernel size, highlighting the need for a scalable approach to address the challenge effectively. Increasing the number of channels provides more degrees of freedom, which may help cover more of the orthogonal convolution space while increasing the kernel size expands the range of spatial features that can be represented.

Research on the complete parametrization of orthogonal convolutions with controlled kernel sizes remains an open question, AOC could benefit from further improvements in this area.

## G. Proofs

### G.1. Proof of the ⊛ property:

Although already shown in previous papers (Li et al., 2019), (Xiao et al., 2018), we provide the proof in the 1D case to help the reader understand the mechanism of the ⊛ operator. Given a vector $x \in \mathbb{R}^{c_{in} \times w}$, and two kernels $\mathbf{A} \in \mathbb{R}^{c_{int} \times c_i \times k_A}$ and $\mathbf{B} \in \mathbb{R}^{c_o \times c_{int} \times k_B}$. We suppose that $\mathbf{A}$ is zero-padded, i.e., $\mathbf{A}_{c,n,i} = 0$ if $i < 0$ or $i \geq k_A$.

*Proof.*

$$y = \mathbf{A} \star x, \text{such as,}$$

$$y_{c,j} = \sum_{n=0}^{c_i-1} \sum_{j'=0}^{k_A-1} \mathbf{A}_{c,n,j'} x_{n,j-j'}$$

$$z = \mathbf{B} \star y = (\mathbf{B} \circledast \mathbf{A}) \star x, \text{such as,}$$

$$
\begin{aligned}
z_{m,l} &= \sum_{c=0}^{c_{int}-1} \sum_{i'=0}^{k_B-1} \mathbf{B}_{m,c,i'} y_{c,l-i'} \\
&= \sum_{c=0}^{c_{int}-1} \sum_{i'=0}^{k_B-1} \mathbf{B}_{m,c,i'} \sum_{k=0}^{c_i-1} \sum_{j'=0}^{k_A-1} \mathbf{A}_{c,k,j'} x_{k,l-i'-j'} \\
&= \sum_{k=0}^{c_i-1} \sum_{j'=0}^{k_A-1} \sum_{c=0}^{c_{int}-1} \sum_{i'=0}^{k_B-1} \mathbf{B}_{m,c,i'} \mathbf{A}_{c,k,j'} x_{k,l-i'-j'} \\
&= \sum_{k=0}^{c_i-1} \sum_{l'=0}^{k_A+k_B-1} \sum_{c=0}^{c_{int}-1} \sum_{i'=0}^{k_B-1} \mathbf{B}_{m,c,i'} \mathbf{A}_{c,k,l'-i'} x_{k,l-l'} && (\text{with } l' = i' + j') \\
&= \sum_{k=0}^{c_i-1} \sum_{l'=0}^{k_A+k_B-1} (\mathbf{B} \circledast \mathbf{A})_{m,k,l'} x_{k,l-l'} && (\text{direct formulation of } (\mathbf{B} \circledast \mathbf{A}) \star x)
\end{aligned}
$$

with thus

$$(\mathbf{B} \circledast \mathbf{A})_{m,n,i} = \sum_{c=0}^{c_{int}-1} \sum_{i'=0}^{k_B-1} \mathbf{B}_{m,c,i'} . \mathbf{A}_{c,n,i-i'}$$

$\square$

## G.2. Recall of $\circledast$ other properties

Two kernels $\mathbf{A}$ and $\mathbf{B}$ are said compatible for the $\mathbf{B} \circledast \mathbf{A}$ operation when the number of input channels of the second convolution $\mathbf{B}$ matches the number of output channels of the first convolution. This condition is denoted as $A \bowtie B$.

We present here several properties of the Block-convolution operator $\circledast$:

**Proposition G.1** (Associativity)**.** *The $\circledast$ operation is associative (given compatible kernels $A \bowtie B$ and $B \bowtie C$):*

$$A \circledast (B \circledast C) = (A \circledast B) \circledast C$$

**Proposition G.2** (Bi-linearity)**.** *Given two convolutions $A$ and $B$ with the same channel sizes, a third convolution $C$ compatible with $A$ and $B$ ($A \bowtie C$, $B \bowtie C$), and two scalars $\lambda_1, \lambda_2 \in \mathbb{R}$:*

$$(\lambda_1 A + \lambda_2 B) \circledast C = \lambda_1 A \circledast C + \lambda_2 B \circledast C$$

**Proposition G.3** (Non-Commutativity)**.** *Even when $A \bowtie B$ and $B \bowtie A$ hold, Block-convolution is not commutative, as convolution composition is generally not commutative:*

$$\mathcal{AB} \neq \mathcal{BA} \implies \mathbf{A} \circledast \mathbf{B} \neq \mathbf{B} \circledast \mathbf{A}$$

## G.3. Proof of the construction of $c \times c \times 1 \times 2$ orthogonal convolution

This proof was already presented in (Li et al., 2019), but we include it here for the sake of completeness. Given a symmetric orthogonal projectors $N \in \mathbb{R}^{c \times c}$ such as:

$$N = N^2 = N^T \quad \text{and} \quad (I - N) = (I - N)^2 = (I - N)^T$$

Considering $N$ and $I - N$ as $\mathbb{R}^{c \times c \times 1 \times 1}$ convolution kernels, we can build a convolution kernel $\mathbf{P} \in \mathbb{R}^{c \times c \times 1 \times 2}$ by:

$$\mathbf{P} = \texttt{stack}([\mathbf{N}, \mathbf{I} - \mathbf{N}], \texttt{axis} = -1)$$

For readability, we will write $\mathbf{P} = [N, I - N]$ as a compact notation for the stacked kernels. We will prove that such a kernel $\mathbf{P}$ defines an orthogonal convolution. Since this convolution has no stride $s = 1$, as stated in Section 2.2, this is equivalent to :

$$\mathbf{P} \circledast \mathbf{P}^T = \mathbf{P}^T \circledast \mathbf{P} = \mathbf{I}$$

*Proof.* We can compute explicitly the resulting kernel for $\mathbf{P} \circledast \mathbf{P}^T$:

$$
\begin{aligned}
\mathbf{P} \circledast \mathbf{P}^T &= \\
&= [N, I - N] \circledast \left[ (I - N)^T, N^T \right] \\
&= \left[ N(I - N)^T, NN^T + (I - N)(I - N)^T, (I - N)N^T \right] \\
&= \left[ N(I - N), N^2 + (I - N)(I - N), (I - N)N \right] \\
&= \left[ N - N^2, N^2 + I - 2N + N^2, N - N^2 \right] \\
&= \left[ N - N, N + I - 2N + 2N^2, N - N \right] \\
&= [0, I, 0]
\end{aligned}
$$

$\square$

The third line is the matrix application of the $\circledast$ with two $1 \times 2$ kernels (Eq. (7)). The following lines are based on the trivial application of the symmetric projector property.

**Construction of symmetric projectors.** Following the construction described in (Li et al., 2019) a symmetric projectors $N \in \mathbb{R}^{c \times c}$ can be constructed based a column-orthogonal matrix $N_0 \in \mathbb{R}^{c \times \frac{c}{2}}$:

$$\text{given } N_0 \in \mathbb{R}^{c \times \frac{c}{2}}, \text{ such that } N_0^T N_0 = I$$

The matrices $N = N_0 N_0^T$ and $I - N$ are a symmetric projectors.

*Proof.* This proof was already presented in (Li et al., 2019), but we include it here for the sake of completeness.

$$
\begin{aligned}
N^2 &= (N_0 N_0^T)(N_0 N_0^T) \\
&= N_0 (N_0^T N_0) N_0^T \\
&= N_0 N_0^T \\
&= N = N^T
\end{aligned}
$$

$$
\begin{aligned}
(I - N)^2 &= (I - N_0 N_0^T)(I - N_0 N_0^T) \\
&= I - 2N_0 N_0^T + (N_0 N_0^T)(N_0 N_0^T) \\
&= I - N_0 N_0^T \\
&= I - N = (I - N)^T
\end{aligned}
$$

$\square$

**G.4. RKO kernel with stride $s = k$ builds an orthogonal convolution**

The RKO method considers a convolution $\text{conv}_K(.,s)$ with a kernel $\mathbf{K} \in \mathbb{R}^{c_o \times c_i \times k \times k}$, built by reshaping an orthogonal matrix $K' \in \mathbb{R}^{c_o \times c_i.k^2}$. The matrix $K'$ can be obtained for instance by the Björck and Bowie orthogonalization scheme (Björck & Bowie, 1971) and verifies $K'K'^T = Id$ for row orthogonality (resp. $K'^T K' = Id$ for column) .

In the general case, the resulting convolution is not orthogonal (Achour et al., 2022). We prove that in the special case where the kernel and the stride are equal, the $\text{conv}_K(.,k)$ with stride $s = k$ is orthogonal.

*Proof.* $\forall x \in \mathbb{R}^{c_i \times h \times w}$, where we suppose that $h, w$ are multiple of $s$, we have:

$$\text{conv}_K(x,k)_{h,i,j} = \sum_{c=0}^{c_i-1} \sum_{i'=0}^{k-1} \sum_{j'=0}^{k-1} K_{h,c,i',j'} x_{c,ik-i',jk-j'}$$

$$= \sum_{c=0}^{c_i k^2-1} K'_{h,c} \bar{x}_{c,i,j}$$

$$= (K'\bar{x})_{h,i,j}$$

where $\bar{x} \in \mathbb{R}^{c_i k^2, \frac{h}{k}, \frac{w}{k}}$ is defined by:

$$\bar{x}_{.,i,j} = \begin{bmatrix} x_{0,ik,jk} \\ x_{0,ik-1,jk} \\ \vdots \\ x_{0,(i-1)k+1,jk} \\ x_{0,ik,jk-1} \\ \vdots \\ x_{1,ik,jk} \\ \vdots \\ x_{c_i-1,(i-1)k+1,(j-1)k+1} \end{bmatrix}$$

Since there is no overlap between the receptive field of each output, $\bar{x} = Px$ where the matrix $P$ is a permutation (similar to an invertible downsampling with factor k), which is by definition orthogonal.

In the case of row orthogonality for $K'$, we have:

$$\text{conv}_K(\text{ConvTranspose}_K(x, \text{stride} = k), k)$$
$$= K'P(K'P)^T x$$
$$= K'PP^T K'^T x \qquad\qquad (\text{P is orthogonal})$$
$$= K'K'^T x \qquad\qquad (\text{K' is orthogonal})$$
$$= x$$

$\square$

**G.5. AOC convolutions are orthogonal**

The construction of our method consists in composing a BCOP kernel $\mathbf{K}_{BCOP} \in \mathbb{R}^{c \times c_i \times k-s \times k-s}$ followed by an RKO kernel $\mathbf{K}_{RKO} \in \mathbb{R}^{c_o \times c \times s \times s}$. As we already proved that each kernel is orthogonal, we know that

$$\mathcal{K}_{BCOP} \mathcal{K}_{BCOP}^T = \mathcal{I} \text{ when } c_i \geq c$$
$$\mathcal{K}_{BCOP}^T \mathcal{K}_{BCOP} = \mathcal{I} \text{ when } c_i \leq c$$

and that

$$(\mathcal{S}_s \mathcal{K}_{RKO})(\mathcal{S}_s \mathcal{K}_{RKO})^T = \mathcal{I} \text{ when } c * s^2 \geq c_o$$
$$(\mathcal{S}_s \mathcal{K}_{RKO})^T (\mathcal{S}_s \mathcal{K}_{RKO}) = \mathcal{I} \text{ when } c * s^2 \leq c_o$$

We prove that with a correct choice of the internal dimension $c$, the strided AOC convolution with kernel $\mathbf{K}_{AOC} = \mathbf{K}_{\text{RKO}} \circledast \mathbf{K}_{\text{BCOP}}$ is orthogonal.

*Proof.* As $c$ (the intermediate number of channels) is a free parameter we can demonstrate that our construction is orthogonal when the convolutions are either both row orthogonal , or both column orthogonal (Proposition 2.3), i.e when:

$$c_i \geq c \text{ and } c \geq \frac{c_o}{s^2} \qquad\qquad \text{(both row orthogonal)}$$

$$\text{or} \quad c_i \leq c \text{ and } c \leq \frac{c_o}{s^2} \qquad\qquad \text{(both column orthogonal)}$$

In the first case, when $c_i \geq \frac{c_o}{s^2}$ The resulting convolution $(\mathcal{S}_s \, \mathcal{K}_{RKO} \, \mathcal{K}_{BCOP})$ is orthogonal for any $c$ such as $\frac{c_o}{s^2} \leq c \leq c_i$:

$$
\begin{aligned}
&(\mathcal{S}_s \, \mathcal{K}_{RKO} \, \mathcal{K}_{BCOP})(\mathcal{S}_s \, \mathcal{K}_{RKO} \, \mathcal{K}_{BCOP})^T \\
=&(\mathcal{S}_s \, \mathcal{K}_{RKO})\mathcal{K}_{BCOP} \, \mathcal{K}_{BCOP}^T(\mathcal{S}_s \, \mathcal{K}_{RKO})^T && (\mathcal{K}_{BCOP} \text{ is row orthogonal since} \quad c \leq c_i) \\
=&(\mathcal{S}_s \, \mathcal{K}_{RKO})(\mathcal{S}_s \, \mathcal{K}_{RKO})^T && (\mathcal{S}_s \, \mathcal{K}_{RKO} \text{ is row orthogonal since} \quad \frac{c_o}{s^2} \leq c) \\
=&\mathcal{I}
\end{aligned}
$$

The maximum possible value for $c$ is $c = c_i$. The second case when $c_i \leq c \leq \frac{c_o}{s^2}$ can be proven the same way where the best choice is $c = \frac{c_o}{s^2}$. $\qquad\square$

### G.6. Transposed orthogonal convolutions

The transposition of a row (resp. column) orthogonal convolution is a column (resp. row) orthogonal. The proof is direct by combining Definition 2.2 and Eq. (10).

*Proof.* Given a row orthogonal convolution defined by $\mathcal{S}_s\mathcal{K}$, i.e such that $(\mathcal{S}_s\mathcal{K})(\mathcal{S}_s\mathcal{K})^T = \mathcal{I}$.

The transposed convolution is defined by $(\mathcal{S}_s\mathcal{K})^T$.

We have $[(\mathcal{S}_s\mathcal{K})^T]^T(\mathcal{S}_s\mathcal{K})^T = (\mathcal{S}_s\mathcal{K})(\mathcal{S}_s\mathcal{K})^T = \mathcal{I}$

The transposed convolution is column orthogonal. $\qquad\square$

The same proof can be applied for column orthogonal convolutions.

### G.7. Grouped orthogonal convolutions

A grouped convolution composed of $g$ kernels $(\mathbf{K_i})_g$ is orthogonal if and only if each individual convolution of kernel $\mathbf{K_i}$ is orthogonal. We will suppose that the convolutions that $c_o \leq c_i$ and without stride $s = 1$. The proof is equivalent for $c_i \leq c_o$ and stride $s > 1$.

*Proof.* The proof uses the fact that the Toeplitz matrix of a grouped convolution is block diagonal.

$$
\mathcal{K} = \begin{bmatrix} \mathcal{K}_0 & 0 & \ldots & 0 \\ 0 & \mathcal{K}_1 & \ldots & 0 \\ \vdots & \vdots & \ldots & \vdots \\ 0 & 0 & \ldots & \mathcal{K}_{g-1} \end{bmatrix}
$$

Suppose that each kernel $\mathcal{K}_i$ is row orthogonal (row is due to the fact that $c_o \leq c_i \Rightarrow \frac{c_o}{g} \leq \frac{c_i}{g}$), i.e. $\mathcal{K}_i\mathcal{K}_i^T = \mathcal{I}$.

$$
\mathcal{K}\mathcal{K}^T = \begin{bmatrix} \mathcal{K}_0 & 0 & \dots & 0 \\ 0 & \mathcal{K}_1 & \dots & 0 \\ \vdots & \vdots & \dots & \vdots \\ 0 & 0 & \dots & \mathcal{K}_{g-1} \end{bmatrix} \times \begin{bmatrix} \mathcal{K}_0^T & 0 & \dots & 0 \\ 0 & \mathcal{K}_1^T & \dots & 0 \\ \vdots & \vdots & \dots & \vdots \\ 0 & 0 & \dots & \mathcal{K}_{g-1}^T \end{bmatrix}
$$

$$
= \begin{bmatrix} \mathcal{K}_0\mathcal{K}_0^T & 0 & \dots & 0 \\ 0 & \mathcal{K}_1\mathcal{K}_1^T & \dots & 0 \\ \vdots & \vdots & \dots & \vdots \\ 0 & 0 & \dots & \mathcal{K}_{g-1}\mathcal{K}_{g-1}^T \end{bmatrix}
$$

$$
= \begin{bmatrix} \mathcal{I} & 0 & \dots & 0 \\ 0 & \mathcal{I} & \dots & 0 \\ \vdots & \vdots & \dots & \vdots \\ 0 & 0 & \dots & \mathcal{I} \end{bmatrix}
$$

$$
= \mathcal{I}
$$

Thus $\mathcal{K}$ is row orthogonal.

Conversely, suppose thet $\mathcal{K}$ is row orthogonal. The condition $\mathcal{K}\mathcal{K}^T = \mathcal{I}$ implies in the matrix multiplication above that $\forall i \in [0, g-1], \mathcal{K}_i\mathcal{K}_i^T = \mathcal{I}$. □

### G.8. Dilated orthogonal convolutions

The proof of equivalence between orthogonality of a dilated convolution and orthogonality of the same convolution without dilation is given in (Su et al., 2022) using the equivalence in the spectral domain. Here we explicit the equivalence of convolution in the spatial domain.

Given a convolution of kernel $K_d$ with a dilation $d$, and an input $x \in \mathbb{R}^{c_i \times h \times w}$. We have:

$$
conv_K(x, dil=d)_{h,i,j} = \sum_{c=0}^{c_i-1} \sum_{i'=0}^{k-1} \sum_{j'=0}^{k-1} K_{h,c,i',j'} x_{c,i-i'd,j-j'd}
$$

$$
= (\mathcal{K}_d\bar{x})_{h,i,j}
$$

Let define $d^2$ reshaped inputs $(X^{m,n})_{m,n \in [0,d-1]} \in \mathbb{R}^{c_i \times \frac{h}{d} \times \frac{w}{d}}$. Where: $X_{c,i,j}^{m,n} = x_{c,m+i*d,n+j*d}$. We also define the vectorized vectors $\overline{X^{m,n}}$.

We thus can write using $ri = i \bmod d, rj = j \bmod d, qi = \lfloor \frac{i}{d} \rfloor, qj = \lfloor \frac{j}{d} \rfloor$:

$$
conv_K(x, dil=d)_{h,i,j} = \sum_{c=0}^{c_i-1} \sum_{i'=0}^{k-1} \sum_{j'=0}^{k-1} K_{h,c,i',j'} X_{c,qi-i',qj-j'}^{ri,rj}
$$

$$
= conv_K(X^{ri,rj}, dil=1)_{h,qi,qj}
$$

$$
= (\mathcal{K}\overline{X^{ri,rj}})_{h,qi,qj}
$$

Thus the convolution with dilation is equivalent to the application of the convolution without dilation applied on a permutation of the input.

### G.9. Conditions under which stride can be directly applied on BCOP

When $c_i > c_o$, stride $\mathcal{S}_s$ can be directly applied on top of the BCOP kernel $\mathcal{K}_{BCOP}$ (row orthogonal) without the need to construct the RKO kernel to handle striding (as explained in Proposition 2.5).

*Proof.* We first prove that $\mathcal{S}_s$ is also row orthogonal: $\mathcal{S}_s \mathcal{S}_s^T = \mathcal{I}$. We can leverage the proof that RKO convolution is orthogonal when $k = s$. As the identity can be built with a RKO kernel. Then we can show that $(\mathcal{S}_s \mathcal{I})(\mathcal{S}_s \mathcal{I})^T = \mathcal{I}$ (Definition 2.2, given that $c_i < \frac{c_o}{s}$). This proves that $\mathcal{S}_s \mathcal{S}_s^T = \mathcal{I}$.

As $\mathcal{S}_s$ and $\mathcal{K}_{BCOP}$ are both row orthogonal, we can apply Proposition 2.3 to show that the strided convolution $\mathcal{S}_s \mathcal{K}_{BCOP}$ is also row orthogonal. The direct strided version of the convolution using the BCOP kernel is thus also row orthogonal. $\qquad\square$

### G.10. Conditions of invertibility of AOC layers

As presented in Appendix A, the invertibility of layers has a direct application in Normalizing Flows. The following recalls that strictly orthogonal convolutions are inherently invertible via the transposed convolution. Given the Eqs. (2) and (3) we have the equivalence:

$$y = \text{conv}_K(x, \text{stride} = s) \iff \bar{y} = \mathcal{S}_s \mathcal{K} \bar{x}$$

If the convolution is strictly orthogonal -i.e. the matrix $\mathcal{S}_s \mathcal{K}$ is square- then Definition 2.2 implies that:

$$(\mathcal{S}_s \mathcal{K})^{-1} = (\mathcal{S}_s \mathcal{K})^T$$

Thus, the inverse of the convolution layer is the transposed convolution:

$$\bar{x} = (\mathcal{S}_s \mathcal{K})^T \bar{y} \iff x = \text{ConvTranspose}_K(y, \text{stride} = s).$$

For strided convolutions, strict orthogonality implies that $c_o = c_i s^2$. This allows to construct Normalizing Flows that reduce the spatial input dimensions by increasing the number of channels. Note that, this is not equivalent to applying an InvertibleDownSampling operation before the convolution since the strided convolution kernel requires $s^2$ fewer parameters. Note that AOC allow to extend invertibilty to grouped and/or dilated convolutions.

## H. Architecture and training details

| Layer Type | Output Shape | Config |
|---|---|---|
| **Input** | [3, 32, 32] | |
| AOC | [256, 16, 16] | stride=2 |
| **Block (Repeated Residuals)** | | |
| Residual Depthwise Block $\times$ 16 | [256, 16, 16] | 3x3 kernel |
| AvgPool2d | [256] | |
| MaxMin | [256] | |
| Fully Connected Layer | [10] | |

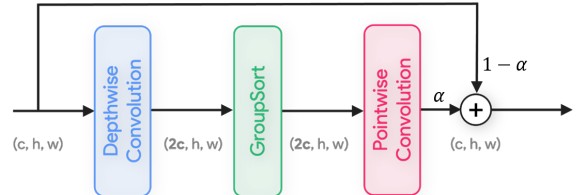

(a) **We can construct complex blocks.** These blocks can reduce the number of parameters of our models, thanks to the flexibility of AOC. Lipschitz continuity is guaranteed when $\alpha \in [0, 1]$.

Figure 5: **Architecture used for CIFAR-10 accurate setting.** This architecture makes use of its flexibility to allow performance with a limited parameter count. **Left,** table of the global network architecture. **Right,** detail of he residual depthwise block.

All hyperparameters of the experiments are detailed in Table 4. These parameters were not heavily tuned: the batch size was set to a large value, but we limited it to 1024 to avoid training for too many epochs. This is due to the fact that robust networks seem to benefit more from more training steps rather than larger batch sizes. Once the batch size was set, the learning rate was tuned the following way: by training multiple times for 500 steps, the learning was increased until the accuracy at the end of these 500 steps was maximized, and the final learning rate was set to the largest values before the metrics plateaus.

For robust training, we must control three elements: accuracy, robustness, and generalization. As noted by (Prach & Lampert, 2024; Béthune et al., 2022), usual scaling laws do not embrace the fact achieving accuracy and robustness simultaneously comes at the price of generalization. Which can be regained by using larger networks, more data augmentation, and longer train times. To control this phenomenon, we can tune the following 3 elements:

- **Loss parameters**: Increasing the margin in the loss function helps improve training robustness, but reduces training

| Hyperparameter | CIFAR-10 Acc. | CIFAR-10 Rob. | ImageNet Acc. | ImageNet Rob. |
|---|---|---|---|---|
| Loss function | Cosine similarity | (Prach & Lampert, 2022) $(m = 1.5\sqrt{2}, \tau = 0.125)$ | Cosine similarity | (Prach & Lampert, 2022) $(m = 1.5\sqrt{2}, \tau = 0.125)$ |
| Optimizer | | ScheduleFree (Defazio et al., 2024) | | |
| Learning rate | $5 \times 10^{-3}$ | $1 \times 10^{-4}$ | $1 \times 10^{-2}$ | $1 \times 10^{-3}$ |
| Batch size | 1024 | 1024 | 512 | 512 |
| Epochs | 150 | 3000 | 90 | 300 |
| Randaugment params | m=6, n=2 | m=6, n=1 | m=7, n=2 | None |
| Random crop params | scale = 0.25 | scale = 0.5 | scale = 0.5 | scale = 0.08 |
| hardware | RTX 3080 x 1 | RTX 4090 x 1 | RTX 4090 x 2 | RTX 4090 x 2 |

Table 4: **Training Hyperparameters for the Four 1-Lipschitz Networks.** "Acc." refers to the high-accuracy setting, and "Rob." refers to the robust setting.

accuracy.

- **Model size**: Increasing the model size generally improves training accuracy.

- **Data augmentation**: Increasing data augmentation reduces training accuracy but can improve validation accuracy, especially when training accuracy is greater than validation accuracy.

The tuning process then works following these steps:

1. start with a given architecture and a low margin $m = 0$

2. increase data augmentation until train accuracy goes below 100%. The model reached its maximum generalization for this margin.

3. increase the margin, increase the model size and number of epochs until it reaches 100% training accuracy.

4. repeat steps 2. and 3. until the desired robustness is achieved. The target margin is $m = \frac{3}{2}\sqrt{2}$ to match the parameters used by (Araujo et al., 2023). In practical contexts, m should be set to $m = \frac{\epsilon\sqrt{2}}{\alpha L}$ where $\epsilon$ is the targeted robustness radius, L the model's Lipschitz constant and $\alpha$ the scaling factor applied on the data (for instance when standard rescaling is performed) the $\sqrt{2}$ factor comes from the equation in Section 3.1.

This explains the drastic difference between standard and robust networks in terms of model sizes and training times.

**Overview of the architectures used** All the architectures used in this paper aims to illustrate that despite its under-parametrization, AOC allows the construction of expressive architectures. All architectures only use AOC convolutions and standard blocks to ensure that performance can be attributed to the method. As underline in (Anil et al., 2019) it is the combination of orthogonal layer and MaxMin activations that permits the construction of networks with a tight estimation of their Lipschitz constant. We then use MaxMin activation in all of our architectures. The reader interested in building 1-Lipschitz networks that use ReLU activations can refer to the work of (Araujo et al., 2023) and (Wang & Manchester, 2023). We describe interactions between AOC and these constructions in Appendix E.

Some of our architectures use skip connections, since those do not construct 1-Lipschitz blocks, we add a learnable factor to correctly ensure the 1-Lipschitzness of the whole network. Given $f_1$ a $l_1$ Lipschitz function and $f_2$ a $l_2$ Lipschitz function, then $f = f_1 + f_2$ is $l_1 + l_2$ Lipschitz. Then the function:

$$y = \frac{\alpha_1 x + f(\alpha_2 x)}{|\alpha_1| + |\alpha_2|}$$

is 1 Lipschitz. This is also true when summing $\alpha_2 f(x)$ instead of $f(\alpha_2 x)$, but the proposed approach allows to control the gradient flowing through $f$. Finally we set $\alpha_1 = 1$ to ensure a better gradient propagation in deep architectures.

| Layer Type | Output Shape | Config |
|---|---|---|
| **Feature Extractor** | | |
| 3 x AOC | [128, 32, 32] | 3x3 kernel |
| AOC | [256, 16, 16] | stride=2 |
| 3 x AOC | [256, 16, 16] | 3x3 kernel |
| AOC | [512, 8, 8] | stride=2 |
| 3 x AOC | [512, 8, 8] | 3x3 kernel |
| AOC | [1024, 4, 4] | stride=2 |
| 4 x AOC | [1024, 4, 4] | |
| Flatten | [8192] | |
| **Fully Connected** | | |
| 4 x OrthogonalDense | [1024] | |
| OrthogonalDense | [10] | |

Table 5: **Architecture of the CIFAR-10 robust network.** unlike other networks, this network has an architecture designed to be similar to the original network from (Li et al., 2019) (with increased width and depth). It follows the original design choices (using circular padding and the absence of global pooling).

**CIFAR-10 (Krizhevsky et al., 2009)**   The CIFAR-10 experiments used three distinct networks, the network tailored for accuracy is detailed in Fig. 5, in this setting zero padding is used, with this padding, the layer is 1-Lipschitz and quasi-orthogonal (orthogonal everywhere except for the images border, see (Delattre et al., 2023; 2024) for details). This architecture features a bloc inspired by (Trockman & Kolter, 2024) which consists of a depthwise convolution that doubles the number of channels, followed by a MaxMin activation function (Anil et al., 2019) to enhance non-linearity, and a pointwise convolution that reduces the number of channels. These layers are encapsulated within a skip connection featuring a learnable factor to ensure a Lipschitz constant of 1. On the other hand, the robust network has an architecture designed to be as similar as possible to the networks used by (Li et al., 2019) which allows to show the impact of scale on the results. The architecture is detailed in Table 5. Finally, the network of the *AOC robust\**, reuses the training recipe of (Hu et al., 2023) with the original convolutions replaced by AOC convolutions (which include downsampling convolutions). The residual connections were removed, and a constant value of 1.15 replaced the scalar factors. This resulted in a network with a tighter Lipschitz constant evaluation, which resulted in non-optimal training with original loss parameters (the training led to networks that were not accurate enough). Hence, the loss parameters were divided by 10 to make the network more accurate.

We tested the robust architecture in an accurate setting and vice versa, we observed that the robust architecture was underperforming in the accurate setting, plateauing at 85% of validation accuracy and 0% of certified robust accuracy. Similarly, the accurate architecture was underperforming in terms of robustness with only 47% of certified robust accuracy and 71% of validation accuracy.

**Imagenet-1K (Deng et al., 2009)**   The ImageNet-1K experiment follows the hyperparameter configuration detailed in Table 4. The two architectures utilized in this study exhibit a high degree of similarity, with their structural differences illustrated in Figure Table 6. Both architectures incorporate the fundamental block described in Section Fig. 5a.

The primary distinction between the accurate and robust settings lies in the architectural modifications aimed at enhancing robustness. Specifically, the robust architecture employs L2-normalized pooling (Boureau et al., 2010) and circular padding, both of which contribute to improved resilience against adversarial perturbations. Furthermore, inspired by (Trockman & Kolter, 2024), the convolutional blocks in the robust model utilize larger kernel sizes to capture a broader spatial context.

### H.1. Scalability experiment

Experiments were conducted on a minimally modified ResNet-34 architecture, chosen for its compatibility with various orthogonal layers and its status as a standard benchmark for ImageNet training. The transition blocks were replaced by a single, strided convolution to maintain simplicity and ensure compatibility with existing orthogonal layers. For each method, we measured the average training and testing times over 100 batches and recorded peak memory consumption. Starting with a batch size of 128, we doubled the batch size incrementally until encountering an out-of-memory error. Each method's

| Layer Type | Output Shape | Config |
|---|---|---|
| Input | [224, 224] | |
| Convolution Layer 1 | [112, 112] | 5x5 kernel |
| Activation Layer 1 | [112, 112] | MaxMin |
| **Block 1** | | |
| Residual Depthwise Block x 3 | [56, 56] | 5x5 kernel |
| Convolution Layer | [28, 28] | 3x3 kernel |
| **Block 2** | | |
| Residual Depthwise Block x 3 | [28, 28] | 5x5 kernel |
| Convolution Layer | [14, 14] | 3x3 kernel |
| **Block 3** | | |
| Residual Depthwise Block x 3 | [14, 14] | 5x5 kernel |
| Convolution Layer | [7, 7] | 3x3 kernel |
| **Block 4** | | |
| Residual Depthwise Block x 3 | [7, 7] | 5x5 kernel |
| L2 Pooling | [1, 1] | 3x3 kernel |
| Flatten | [2048] | |
| Fully Connected Layer | [1000] | |

Table 6: **Summary of our architecture used on Imagenet-1K.**

performance was compared to a standard convolution baseline, with results reported as overhead percentages.

To ensure a fair comparison, the hyperparameters of SOC and Cayley were set to their default values. We adjusted the number of Björck iterations to the same values for BCOP and AOC. While originally set to 20 iterations, our unit testing scheme (see Appendix D) shows that 12 iterations are sufficient to ensure a stable rank (Sanyal et al., 2020) of 99.9% of the full rank. Finally, standard `Conv2D` is used with circular padding to evaluate the overhead induced by our parametrization rather than the overhead induced by the padding.

We did not include non-orthogonal methods such as AOL, SLL, CPL and RKO (Prach et al., 2024; Araujo et al., 2023; Meunier et al., 2022; Serrurier et al., 2021) as those are expected to be faster than AOC that add a stronger constraint. The continuum between orthogonal and non-orthogonal methods could be explored by reducing the number of Björck and Bowie iterations. However, the correct way to explore this should be with an experiment that grasp the 3 main aspects of those methods: speed (time per batch), trainability (number of batches to reach certain accuracy) and performance (final accuracy/robustness). This is beyond the scope of this work, and the work of (Prach et al., 2024) is a relevant track to explore this question.

## I. Reproducing the results from BCOP paper

As mentioned earlier, AOC is built on top of components like BCOP and RKO. Hence, any accuracy gain must come from the fact that our implementation allows larger networks and more training steps within the same compute budget. To illustrate this, we reproduced the baseline results from (Li et al., 2019) and switched the implementation to ours. Results are shown in Table 7. We can observe a notable difference in performance between the original implementation and ours. This is due to the difference in the parametrization of strided convolutions: the original paper uses invertible downsampling to emulate striding followed by a standard convolution whose number of input channels is multiplied by $s^2$, such convolution has a $s^2 \times$ more parameters than the proposed AOC strided convolution. This is accentuated by the specific small architecture used in BCOP paper (and reproduced in this experiment) which has 84% of the convolutional layers parameters that are located in strided convolutions. Also, invertible down-sampling has a $2 \times 2$ receptive field, which increases the global convolution's receptive field. However, when we remove the optimization depicted in Fig. 2c and only resort to the parametrization described in Fig. 1, the increase in the number of parameters results in improved results close to the results of the original paper. This observation is non-trivial since this modification is not equivalent to the original implementation: instead of parametrizing a $c_o \times 4 c_i \times k \times k$ AOC parameterized one $\max(c_i, \frac{c_o}{s^2}) \times c_i \times k - s + 1 \times k - s + 1$ and one $c_o \times \max(c_i, \frac{c_o}{s^2}) \times s \times s$ kernels which are much smaller.

| Models | Accuracy | Provable Accuracy $\epsilon = \frac{36}{255}$ |
|---|---|---|
| BCOP - Small net (original seeting) | 72.2 | 58.26 |
| AOC - Small net (all optimizations) | 62.3 | 49.18 |
| AOC - Small net (opt in Fig. 2c removed) | 71.8 | 58.25 |
| AOC - Large net (all optimizations) | 74.0 | 64.33 |

Table 7: **Mitigating AOC limitations in small scale setting**: as AOC uses less parameters for strided convolutions, this can impact its expressive power in small scale setting. However, by removing the optimization in Fig. 2c this increase the number of parameters enough to correct this issue. Results from Table 2 added for reference.

We recall that all the results of this paper were obtained with the optimization in Fig. 2c. We evaluated the unoptimized version in the same context as in Table 3 and found, for a batch size of 512, a slowdown of $1.21\times$ in train time (instead of $1.13\times$) and no notable increase in train memory consumption.

Finally, it is worth noting that the optimization depicted in Fig. 2c should not affect the expressive power if we were able to parameterize the complete set of orthogonal convolutions.

