# OpenReview forum: "An Adaptive Orthogonal Convolution Scheme for Efficient and Flexible CNN Architectures"
_ICML.cc/2025/Conference — ICML 2025 poster_

### Official Review · Reviewer_BhA8 · 2025-03-03

**Overall Recommendation:** 4

**Summary:**

The paper considers the problem of constructing convolutions which correspond to orthogonal operators. While the general approach is based on BCOP (Li et al. 2019), the authors extend this framework to certain variants of convolutions like stride, dilations, grouping and transposing and consider some aspects for reducing the computational cost. A numerical example for classification on CIFAR10 shows that these extensions can improve the expressiveness of orthogonal convolutional neural networks.

**Claims And Evidence:**

The claims are clear are sufficiently supported.

**Essential References Not Discussed:**

Many properties of orthogonal convolutions (matrix algebra/manifold property, orthogonal projections, an additional training algorithm etc.) were discussed in the paper:

Hertrich et al. "Convolutional proximal neural networks and plug-and-play algorithms", Linear Algebra and its Applications, 2021

This is highly relevant to the paper and should be discussed.

**Experimental Designs Or Analyses:**

The numerical setup is generally feasible to validate the claims of the paper. However, the description in the paper should be adjusted:

- What is accurate and robust AOC? Are only the architectures different or also the training? Can you also construct an accurate/robust BCOP?

- Please include some SOTA non-orthogonal network as a reference to show the gap between orthogonal and non-orthogonal networks

- It seems like hyper-parameters and precise architectures play a massive role for the results. It would be good to compare the different convolution approaches with minimal differences in the rest of the hyperparameters...

**Methods And Evaluation Criteria:**

Methods and evaluation are feasible.

**Other Comments Or Suggestions:**

Please proofread the paper again and correct typos. Some examples:

- line 196/197 left: missing spaces before references

- line 212/213: "convolutions kernel" -> "convolution kernels"

- line 1587: broken reference "Table ??"

**Other Strengths And Weaknesses:**

The main construction of the orthogonal convolutions in this paper is taken from the literature (Li et. al 2019, Xiao et al. 2018). However, the authors take care of all the variants of convolutions, which are used in literature. This includes stride, dilations, grouping and computational efficiency (and also transposed convolutions, but this part is a bit trivial). I consider this as a solid contribution. Additionally, the authors provide a PyTorch library for implementing their and several other orthogonal or 1-Lipschitz networks. Some weaknesses are listed below.

In summary, I would see the paper above the acceptance threshold, but I would wish that the authors reconsider their wording to describe their contribution more appropriately.

## Weaknesses

### Inaccurate names, abstract, introduction

- I find the name of the method (adaptive orthogonal convolutions) a bit odd. Under adaptivity I would understand that the architecture adapts to the data/training process or whatever.

- The authors overstate (or rather misstate) their contributions a bit. In abstract and introduction, the authors write, they would introduce a new method. Instead they should rather present their contribution as it is: The introduction of stride, dilation, grouping etc. to BCOP.

### Other

- It would be useful to give a bit more intuition on some of the claims. For instance, Prop 2.4 basically considers the setting where each pixel in the input image is covered exactly once from each kernel. This is one of the main reasons, why the orthogonality condition simplifies in this case.

- Proposition 2.7 is trivial. It basically states that the transposed matrix of a matrix with orthogonal columns has orthogonal rows...

**Questions For Authors:**

see above.

# After Rebuttal

Many thanks to the authors for answering my questions and comments. I increased the score from 3 to 4.

**Relation To Broader Scientific Literature:**

The general construction of orthogonal convolutions is taken from (Li et. al 2019, Xiao et al. 2018). The contributions of the paper are the extension to common variants of convolutions like stride, dilation, grouping.

**Theoretical Claims:**

I only read the proofs, which are not stated to be contained already in the literature. The remaining ones seem to be correct, even though their are quite simple (the strengths of the paper are more in the modelling/conceptual part than in proving complicated theorems).

---

> ### Author Rebuttal · Authors · 2025-03-29
>
> We sincerely appreciate the detailed proof verification, the interest you've shown in our paper, and the evaluation towards acceptance. We answer the main weaknesses raised in your review, and hope this can help you to increase your score.
>
> About experiments
> ---------
>
>  > What is accurate and robust AOC? Are only the architectures different or also the training?
>
> The architectures, detailed in Appendix H, play a role in the accuracy/robustness tradeoff, but the main factor remains the loss:
> - The “robust” setting uses the scaled cross-entropy with margin (Prach et al.) to promote robustness.
> - The “accurate” setting uses Cosine similarity, chosen for its scale invariance. (Bethune 2021) showed the standard cross-entropy contains a term that promotes robustness (as softmax is not scale invariant).
>
> This will be clarified in the main part of the manuscript (sect 3.1 l383).
>
>
> > Can you also construct an accurate/robust BCOP?
>
> The “BCOP accurate”  leads to the same accuracy as the “AOC accurate” on Cifar-10, since as stated l405 “AOC does not improve the expressiveness of its original building blocks (namely BCOP)”, and was thus not reported. The BCOP robust on Cifar10 is equivalent to the first line of Table 2. However, note that the BCOP method cannot scale on the Imagenet dataset.
>
> > Please include some SOTA non-orthogonal network as a reference to show the gap between orthogonal and non-orthogonal networks
>
> Table 2 includes some SOTA non-orthogonal networks, such as AOL, SLL, or Li-Resnet, that are non-orthogonal.
>
>
> > It would be good to compare the different convolution approaches with minimal differences in the rest of the hyperparameters…
>
> We made the choice to compare the published version for each method since the optimal architecture can be different. An empirical comparison of previous methods was already done by (Pratch et al.). The authors chose to use a fixed budget of 24 GPU hours, Since AOC is much faster than BCOP, It is expected to outperform BCOP in their comparison.
>
>
>
> About novelty and name of the method
> -----------
>
> > The authors overstate (or rather misstate) their contributions a bit. In abstract and introduction, the authors write, they would introduce a new method. Instead they should rather present their contribution as it is: The introduction of stride, dilation, grouping etc. to BCOP
>
> The abstract indicates that “AOC  [is] a scalable method for constructing orthogonal convolutions, effectively
> overcoming these limitations [i.e. strides, dilations, group, convolutions, and transposed convolutions]”, we don’t understand what in this sentence “overstates” the contribution? Could you please clarify?
>
> In the introduction, we state clearly the novelty and difference with BCOP  in Table 1. The bullet points in l82-l102 list the properties AOC achieves **simultaneously**.
>
> Also, besides the kernel construction, our contribution is 3 fold:
> - The method itself. Despite the method's apparent simplicity, the work to achieve this is non-trivial: as discussed in the paper, the construction requires multiple criteria to be orthogonal, and its proof requires numerous results to be complete.
> - A mathematical framework that allows one to navigate between three aspects of the convolutions seamlessly: its kernel tensor, its Toeplitz matrix, and its pytorch code. This makes many results simple to prove and extends beyond AOC (Appendix E shows how we can improve SLL, SOC, and Sandwich in nontrivial ways)
> - An efficient implementation that makes the usual slowdown negligible. That is packaged in an open-source library. This library also centralizes and improves multiple existing methods (as stated in Appendix E)
>
>
> > name of the method (adaptive orthogonal convolutions) a bit odd
>
> We understand the concern of the reviewer , we use the word ‘adaptive’ in the sense that it can be adjusted to the layer modern features. We were thinking about Flexible Orth Conv , but the acronyms is often understood as Free-Of-Charge
>
> Others
> ------
>
> >   give a bit more intuition on some of the claims. For instance, Prop 2.4
>
> We appreciate the reviewer’s suggestion to provide more intuition. We will provide when possible in the main paper an intuition of the proof.
>
> > Proposition 2.7 is trivial. It basically states that the transposed matrix of a matrix with orthogonal columns has orthogonal rows
>
> We acknowledge that the mathematical framework we introduced makes the proof of this proposition trivial. However, it has implications regarding practical implementation (direct use of torch.nn.conv_transpose_2d) and orthogonality under stride, groups and dilation. For instance, as indicated l251, transposition of stride convolution could be used in upsampling layers of U-Nets.
>
> Reference
> ------
>
> We would like to thank the reviewer for the pointer on (Hertrich et al.), which we missed as we focused on reparametrization methods. We totally agree with its relevance.

---

> > ### Comment · Reviewer_BhA8 · 2025-04-01
> >
> > Many thanks for your detailed answers. I have some additional comments/questions/clarification for my original comments below. I am happy to reconsider my score afterwards.
> >
> > > The abstract indicates that “AOC [is] a scalable method for constructing orthogonal convolutions, effectively overcoming these limitations [i.e. strides, dilations, group, convolutions, and transposed convolutions]”, we don’t understand what in this sentence “overstates” the contribution? Could you please clarify?
> >
> > My critics about this point is the following: The paper mainly proposes to combine BCOP and RKO in a clever way, but does not introduce any new building blocks by itself. But the sentence "we introduce AOC (Adaptative Orthogonal Convolution), a scalable method..." complete erases this history. To provide a suggestion, I would be fine with something like
> >
> > > "We introduce AOC (Adaptative Orthogonal Convolution), which combines BCOP and RKO. In this way, we obtain a scalable method..."
> >
> > A similar addition about the close relyance on BKOP/RKO is necessary in the contributions and conclusions part.
> >
> > To be clear: My concern is not about novelty/contribution of the paper itself, but rather that it heavily builds on BCOP/RKO without acknowledging this fact properly in abstract, contributions and conclusions.
> >
> > As a side note: I agree that I should have mentioned the code library in my original review. I added it in the strenghts and weaknesses part.
> >
> > Regarding:
> > > Table 2 includes some SOTA non-orthogonal networks, such as AOL, SLL, or Li-Resnet, that are non-orthogonal.
> >
> > In this case I would suggest to mark in Table 2 which approaches are orthogonal and which not. Additionally, these approaches are still 1-Lipschitz right? My original intention was to include a comparison what is possible with a network without such constraints (which can be very restricting).

---

> > > ### Author Response · Authors · 2025-04-03
> > >
> > > Thank you for clarifying your point, it can be clarified as mentioned to better contextualize our contributions in abstract, introduction and conclusion.
> > >
> > > About Table 2: Yes, all methods train Lipschitz-bounded networks to ensure certified accuracy. Some methods don't train orthogonal networks, which can be inferred from Table 1. Since some approaches don't propose new layers, the orthogonality of those is not stated explicitly. In order to provide a better picture here is the list of all networks from Table 2 with this information:
> > >
> > > | method      | orthogonal |
> > > |-------------|------------|
> > > | BCOP        | yes        |
> > > | GloRo       | no         |
> > > | Local-Lip-B | no         |
> > > | Cayley      | yes        |
> > > | SOC         | yes        |
> > > | CPL         | no         |
> > > | AOL         | no         |
> > > | SLL         | no         |
> > > | Li-Resnet   | no         |
> > > | AOC         | yes         |

---

### Official Review · Reviewer_HsiT · 2025-03-09

**Overall Recommendation:** 2

**Summary:**

The paper proposed a new method to design scalable and versatile orthogonal convolutional layers.

This layer allows scaling further architecture composed of orthogonal layers; indeed, previous orthogonal layers lack common features of regular convolutional layers (strides, dilations, group convolutions, etc).

The so-called AOC layer encompasses those new features, theoretical demonstrations are provided to support those claims. Also, AOC relies on a faster implementation of block convolution, which makes the computational overhead close to a vanilla convolutional layer.

The utility of AOC layers is demonstrated on certified adversarial robustness tasks and computational time comparisons across various methods.

**Claims And Evidence:**

All the claims in the paper are backed by clear and convincing evidence.

**Essential References Not Discussed:**

All references are correct, but maybe adding a reference on (Salman et al., 2019), which cast randomized smoothing network as Lipschitz network.

**Experimental Designs Or Analyses:**

Experimental designs are valid for the adversarial robustness section, but more experience on other tasks is required (with application in normalizing flows and GANs as promised in the introduction or other tasks for orthogonal architecture).

**Methods And Evaluation Criteria:**

In the abstract, it is claimed that orthogonal convolutional layers are important blocks for adversarial robustness, normalizing flows, GANs, and Lipschitz-constrained models.
However, in the experiments, only adversarial robustness tasks are considered. More diverse applications are welcome to demonstrate the use of  AOC and orthogonal property.

**Other Comments Or Suggestions:**

l.20: a dot is missing "finvertible residual networks (Behrmann et al., 2019) Additionally"

Table 1: BCOP (Li et al., 2019) not (Singal & Feizi, 2021)

l.1465 : there is a dot alone.

**Other Strengths And Weaknesses:**

# Strengths

## Reduced computational overhead

The implementation of AOC has been well optimized.

## Training stability

The proposed method seems to achieve competitive accuracy on ImageNet (68.2%) without any batch norm.

## Code base
The authors provided a code base that is clear and complete.


# Weakness

## Introduction

l.45:  "with 1-Lipschitz networks being a prime candidate (Anil et al., 2019) – an approach that requires the use of orthogonal layers".
$1$-Lipschitz network can be designed through the product upper bound and Lipschitz layers one particular case of 1-Lipschitz layer is orthogonal layers **but it is not the only solution** (Meunier et al.; 2022, Araujo et al.2023; Wang et al., 2023; Hu et al., 2023).

l.27: "Wasserstein GANs (WGANs)  [...] orthogonality in both the discriminator and generator (Miyato et al., 2018; [...]"
Maybe I did not understand well, but Miyato et al., 2018 proposed a spectral normalization technique and not orthogonalization.
Also, the Lipschitz constraint is only enforced on the discriminator in WGANs.

l.38: "Efficient orthogonalization of these structured matrices has theoretical importance, affecting generalization (Bethune et al., 2022)"
I don't see the **direct link** with efficient orthogonalization for layer and the work of Bethune et al., 2022.
The lipschitz constant of the neural network divided by the margin has been linked to reducing the generalization gap (Bartlett et al., 2017), but those considerations are independent of the implementation of Lipschitz layers and, in particular, orthogonal ones.

l.87: "Relaxed orthogonality approaches. In some cases, strict
orthogonality is relaxed to mitigate vanishing gradients,
avoiding the computational demands of full orthogonaliza-
tion (Prach & Lampert, 2022; Meunier et al., 2022; Araujo
et al., 2023)"  here the approaches of (Meunier et al., 2022; Araujo
et al., 2023) do not rely on orthogonality and relaxed orthogonality.

Overall, normalizing flows, GANs are much discussed in the introduction but do not appear later in the paper, and their mentioning in the Appendix is too superficial and lacks contribution to motivate AOC.

Also, the introduction feels a bit like a related work section, particularly the part on orthogonal convolutions, there is a back and forth between intro - related work - intro.

## Section 2

l.169: "For the second kernel, RKO
(Serrurier et al., 2021) is the only viable option, as all other
methods depend on stride emulation." RKO has been introduced in prior work by (Li et al., 2019), also redundant with l.219.

l.171 : "For standard orthogonal convolutions with a fixed kernel size, we rely on three
main works (Xiao et al., 2018), (Li et al., 2019), and (Su
et al., 2022)." Redundant with  l.133.

Prop. 2.4 : It is row /column orthogonal as there is stride (not $\ell_2$norm preserving) and not orthogonal $Q^\top Q = Q Q^\top = I$ ($\ell_2$norm preserving).




## Novelty

It should be stated clearly that the proposed AOC is an incremental update of BCOP and RKO (Li et al., 2019).
The contributions related to **Orthogonal**, **Explicit** are already developed by (Li et al., 2019) and cannot be claimed.

Overall, the paper is a well-written review of orthogonal convolutional layers.
However, I don't think combining two existing methods (RKO and BCOP) is a sufficiently novel contribution.

## Experiments

1) in table 2. Why not put ResNet18, which also has 0 % provable accuracy and compare with AOC ?


l.380 : "The overall Lipschitz constant of a sequence of layers is typically
estimated as the product of the individual layer constants;
however, this bound is often loose, and computing the ex-
act constant is NP-hard (Virmaux & Scaman, 2018)" in introduction.

l.365: for certificate equation give reference.

# References

Bartlett, P. L., Foster, D. J., & Telgarsky, M. J. (2017). Spectrally-normalized margin bounds for neural networks. Advances in Neural Information Processing Systems (NeurIPS).

**Questions For Authors:**

## Questions

### Q0)
In the introduction, there is confusion between the Lipschitz constraint and the orthogonal constraint. Please modify it to better reflect that orthgonality is a subset of the Lipschitz constraint.
Also, in section 2, there is confusion between row/column orthogonal and orthogonal.

### Q1)
Can you confirm that you do not use any batch norm to obtain accuracy on ImageNet with AOC architecture (68.2%)? I think it should be discussed more in the paper as it is a good contribution. A related work on optimization without batch norm, gradient clipping could be a good add to the paper.

### Q2)
The proposed method unlocks scalability with the AOC layer. Can you try to scale the capacity of models beyond what has been proposed yet for orthogonal CNNs (even Lipschitz CNNs)? In terms of depth and width.
The scaling law of robustness states that Lipschitz networks require a lot more parameters than regular networks to scale (Bubeck et al., 2021).

### Q3)
Regarding invertible networks for NF, is there a particular use of AOC possible to define the inverse of an orthogonal convolutional layer?
Is there a link with transposed orthogonal convolution?

### Q4)
In experiments, "This is especially notable as our experiments did
not use techniques such as last layer normalization(Singla
& Feizi, 2022), certificate regularization (Singla & Feizi,
2021b), or DDPM augmentation (Hu et al., 2023)." Can you test your method with those techniques to have a fair comparison with LiResNet and SOC? Reported certified accuracy on CIFAR-10 $64.3$% is way below the LiResNet SOTA ($78.1$%).

### Q5)
Can you provide tasks for normalizing flows and GANs or other applications of orthogonal networks that would highlight the AOC layer?

I am willing to increase my score if my concerns were to be addressed.

## References

Sebastien Bubeck and Mark Sellke. A Universal Law of Robustness via Isoperimetry.  Advances in Neural Information Processing Systems, 2021.

**Relation To Broader Scientific Literature:**

Certified robustness with Lispchitz networks still lags behind the empirical robustness obtained with the adversarial training method or randomized smoothing. Thus, work pushing boundaries in this direction is important.

**Theoretical Claims:**

I checked the correctness of all proofs, and they seem correct.

---

> ### Author Rebuttal · Authors · 2025-03-29
>
> We would like to thank the reviewer for his comprehensive review. We sincerely appreciate the effort put into it and hope our response will be at the same level. We organized our answers into 4 sections that cover the various questions and remarks.
>
> About the experiments:
> ------------------------------
>
> - Q5) GANs and normalizing flows: We also believe that AOC can be impactful for GANs/NF; however, the training competitive GAN/NF deserves its own 9 pages and are let for future works as indicated in the conclusion. In L375, we explained our choices to evaluate AOC. These experiments were done to support two claims: 1. AOC is expressive, and 2. AOC is the most viable option for training large-scale orthogonal CNNs.
> - Q2) Thank you for the relevant reference. The scalability of AOC is first demonstrated In Table 2 as AOC is the only method to **successfully train an *orthogonal* CNN on Imagenet-1K**. Moreover Table 3 shows the low memory consumption and low overhead which are key for further scaling of orthogonal networks.
> - Q4) In Table 2, we took the number from (Hu et al. 2023, “A recipe for…”) instead of (Hu et al.2023, “Unlocking …”). We apologize for the issue, and these numbers will be updated. **AOC tested with similar techniques as (Hu et al. 2023) achieves 74.85% of certified robust accuracy with a network that has half the width of LiResnet** (the width was set to 256 instead of 512 to obtain results within the rebuttal period).
> - Why we did not put ResNet18 in Table 2: We chose to display only Lipschitz-constrained networks (i.e. that offer robustness certificates) to avoid losing the reader.
>
> About novelty:
> -------------------
>
> We state clearly the novelty and difference with BCOP in Table 1. The bullet points in l82-l102 list the properties AOC achieves **simultaneously**.
>
> Our contributions are 3 fold:
> 1. The method itself. Despite the method's apparent simplicity, the work to achieve this is non-trivial: as discussed in the paper, the construction requires multiple criteria to be orthogonal (and a special attention on the choice of channel dimensions and the order of convolution composition), and its proof requires multiple results to be complete.
> 2. A mathematical framework that allows one to navigate between three aspects of the convolutions seamlessly: its kernel tensor, its Toeplitz matrix, and its pytorch code. This makes many results simple to prove and extends beyond AOC (Appendix E shows how we can improve SLL, SOC, and Sandwich in nontrivial ways)
> 3. An efficient implementation that makes the usual slowdown negligible. That is packaged in an open-source library. This library also centralizes and improves multiple existing methods (as stated in Appendix E)
>
> About the writing:
> ----------------------
>
>  - Q0.a: Lipschitz/orthogonal constraints: We agree with suggestion to move the L380 in the introduction  and will use it to clarify the point.
>  - Q0.b: For ease of reading, we use “orthogonal” to refer to “row/column orthogonal” depending on the shape. We state “row/column” when this information is important. It will be clarified in L161.
>  - L45: orthogonal refers to (Anil et al. 2019) and not to the global 1-Lip approach. This will be clarified by moving “(Anil et al. 2019) an approach that requires the use of orthogonal layers.” to line 12.
>  - l27: (Miyato et al., 2018) will be moved to l26.
>  - L38: We cited this theoretical paper to balance the citations of (Wang et al., 2020; Qi et al., 2020) that are more empirical. We agree that the link is indirect since the generalization bounds assume 1-Lipschitz networks constructed using the product bound and not orthogonal layers. It can be removed if the reviewer considers this misleading.
>  - L87: We agree that “mitigating vanishing gradient” is a more appropriate title for this paragraph. Thanks for the suggestion.
>  - L365: We cited (Anil et al. 2019) for its direct application to certifiable robustness, but we agree with the relevance of (Bartlett et al. 2017).
>
> Questions:
> ---------------
>
> Q1 [Batchnorm]:
> Yes, all our networks are trained without batch normalization (in the line of (Xiao et al., 2018)). This is especially true for our imagenet results. We agree this is We agree this is important to highlight in the paper.
>
> Q3 [invertible convolution layer]: In our framework, this demonstration becomes easy. Given
>
> $ y = \text{conv}_{K}\text{(x, stride=s)}  \quad \Longleftrightarrow \quad  \bar{y}  = (\mathcal{S}_s\mathcal{K})\bar{x} \quad  \text{(Eq.2-3)}$
>
> If the convolution is orthogonal (not row/column orthogonal), we have
>
> $(\mathcal{S}_s\mathcal{K})^{-1} = (\mathcal{S}_s\mathcal{K})^T \quad \text{(Def.2.2)}$ and then,
>
> $ \bar{x} = (\mathcal{S}_s \mathcal{K})^T \bar{y} \quad \Longleftrightarrow \quad x = \text{ConvTranspose}_K\text{(y, stride=s)} \quad \text{(Def 2.6)}$

---

> > ### Comment · Reviewer_HsiT · 2025-04-06
> >
> > Thank you for the detailed rebuttal and clarifications. While I appreciate the engineering effort and clear exposition, I still believe the contribution is too incremental and lacks novelty, mainly combining existing methods (BCOP and RKO) from the same paper (Li et al. 2019). I will keep my score unchanged.

---

> > > ### Author Response · Authors · 2025-04-09
> > >
> > > We understand the reviewer's personal point of view but would like to clarify that having a method that is simple to understand and apply does not mean that such a method is simple to find (or to prove). We also want to highlight that the applications of our framework go beyond the sole construction of AOC. Appendix E, where we improve SLL, SOC, and sandwich layers **in nontrivial ways**, is a good example of this.

---

### Official Review · Reviewer_dhMd · 2025-03-10

**Overall Recommendation:** 4

**Summary:**

This paper explores research on orthogonal convolutional layers and introduces AOC, a scalable method for constructing orthogonal convolutions. The authors provide a detailed introduction and implementation of their methodology, including the construction of strided, transposed, grouped, and dilated orthogonal convolutions using AOC. Experimental results demonstrate the effectiveness of the proposed framework.

**update after rebuttal**

Thank you for your rebuttal. I will keep my score unchanged and remain positive about this paper.

**Claims And Evidence:**

Please refer to Strengths And Weaknesses.

**Essential References Not Discussed:**

Please refer to Strengths And Weaknesses.

**Experimental Designs Or Analyses:**

Please refer to Strengths And Weaknesses.

**Methods And Evaluation Criteria:**

Please refer to Strengths And Weaknesses.

**Other Comments Or Suggestions:**

Please refer to Strengths And Weaknesses.

**Other Strengths And Weaknesses:**

**Strength**
1. The research on Adaptive Orthogonal Convolution is both intriguing and highly practical. It makes a significant contribution to neural network architecture development and holds great potential for the research community.
2. The paper is well-structured and easy to follow.
3. The implementation of AOC is thorough, and the experimental results effectively demonstrate the significance of the proposed method.

**Weakness**
1. I guess there may be some criticisms regarding novelty (possibly?), as the paper leans more toward an implementation-oriented approach from my perspective. But for me, it's basically good enough. And it does provide additional information/knowledge to me. If the authors could offer a more in-depth and insightful analysis to meet the taste of academic community, it would further enhance the paper’s impact.

**Questions For Authors:**

Please refer to Strengths And Weaknesses.

**Relation To Broader Scientific Literature:**

Please refer to Strengths And Weaknesses.

**Theoretical Claims:**

Please refer to Strengths And Weaknesses.

---

> ### Author Rebuttal · Authors · 2025-03-29
>
> We sincerely appreciate your thoughtful evaluation and the interest you've shown in our paper, particularly regarding its practical contributions.
>
> In addition to the implementation-oriented aspects that you seem to highly appreciate, we would like to highlight the theoretical novelty of our work: The proposed mathematical framework provides a unified perspective over three key aspects of convolutions that allows the theoretical proofs of the orthogonality for five distinct types of convolution operations (classic, dilated, strided, group, transpose).
>
> Besides, we would like to emphasize that, due to the constraint of orthogonal convolution (such as the existence described in Achour et al. 2022, or proposition 2.3), the choice of channel dimensions and the order of convolution composition are both non-trivial and essential to ensure orthogonality (which is another contribution of this paper).
>
> We hope this further clarifies the depth and originality of our contributions and reinforces your positive view of the paper’s potential impact.

---

### Official Review · Reviewer_yVT2 · 2025-03-13

**Overall Recommendation:** 3

**Summary:**

This paper purposes a orthogonal CNN structure. Orthogonal convolution has been shown success in BCOP and RKO. Utilizing the property of the product of orthogonal matrices are still orthogonal (prop 2.3), this paper invents AOC in Eq. 8, which is the product of RKO and BCOP. A parallel computing technique (fig 2) is also proposed to significantly increase the performance (Tab 3). Comparing to existing works, this algorithm is also easily applied to striding, dilating, etc. I am not very familiar with the area so please point out when anything I say is incorrect.

1. It seems to me the main method AOC is to simply attach a RKO layer to a BCOP layer. Although this idea seems straight forward, it has good performance shown in Tab 2. Do you have an intuition why it is so good?

2. On CIfar 10, why the accuracies shown in Tab. 2 are so low? I imagine modern CNNs with several million parameters will easily reach >90% accuracy.

3. Why AOC provable accuracy is 00.0 in Tab. 2? Is that a typo?

4. Tiny suggestion: do you think it would be better if containing the number of parameters in Table 3 would be better? It seems although AOC has less number of parameters comparing to resnet (53.1M vs 86.0 M as shown in Tab. 2 for IN-1K), it requires longer time during inference in Tab. 3. Am I missing something?

Currently I would keep my rating at weak accept and I will keep trying to better understand this work during the discussion period. Looking forward to the reply from the authors!

**Claims And Evidence:**

-

**Essential References Not Discussed:**

-

**Experimental Designs Or Analyses:**

-

**Methods And Evaluation Criteria:**

-

**Other Comments Or Suggestions:**

-

**Other Strengths And Weaknesses:**

-

**Questions For Authors:**

-

**Relation To Broader Scientific Literature:**

-

**Theoretical Claims:**

-

---

> ### Author Rebuttal · Authors · 2025-03-29
>
> We want to thank the reviewer for his review. We will provide additional information that we hope you to find relevant.
>
> **About 1.** While AOC does not improve the expressiveness of its original building blocks, its flexibility unlocks the construction of complex blocks as depicted in Fig. 5a. Given a fixed compute budget, AOC allows the training of larger architectures for more steps, (thanks to our fast implementation, which is also our contribution). This ultimately leads to improved performances.
>
> **About 2&3.** Certifiable robustness gives guarantees for any possible adversarial attack. However, multiple works have shown a tradeoff between robustness and accuracy. This tradeoff is responsible for the low accuracies found in this field. “AOC accurate” lines show that our networks are as expressive as unconstrained networks when the training does not optimize certified robustness. This leads to 0.0% certified robustness (even if empirical attacks may not achieve this 0% score).
>
> **About 4.** Although the authors dubbed their network as “liresnets” their architecture significantly differs from the usual resnets (as our network). Given these differences, we chose to stick to a known architecture (Resnet34) in Table 3 to ensure a fair comparison.
>
> We hope that these points will convince you of the paper's potential.

---

> > ### Comment · Reviewer_yVT2 · 2025-04-09
> >
> > i deeply thank the authors for the nice paper and the effort from AC for holding such a great conference. I will keep my rating for the following reasons.
> >
> > I am partially convinced for 1.
> >
> > For 2, 3, 4, I am not convinced that `tradeoff between robustness and accuracy' leads to low accuracy. To me, I still think the performance on CIFAR is too low. For example, the original paper of ResNet https://arxiv.org/pdf/1512.03385 shows CIFAR10 with >90% accuracy with <1M parameters.
> >
> > Best wishes
> >
> > yVT2

---

> > > ### Author Response · Authors · 2025-04-09
> > >
> > > Thank you for your remark about the trade-off between accuracy and robustness; one could think that higher robustness implies a higher accuracy. However, the work of (Béthune et al. 2022) showed that the minimization problem to solve robust training involves such a trade-off (in Fig 2 of their paper); this is confirmed empirically by (Prach et al. 2024) in Fig 4.
> > >
> > > The literature usually reports only the best robust accuracy, as shown in our Table 2, which hides this tradeoff.
> > >
> > >
> > > (Béthune et al. 2022) Béthune, Louis, et al. "Pay attention to your loss: understanding misconceptions about lipschitz neural networks." Advances in Neural Information Processing Systems 35 (2022): 20077-20091.
> > >
> > > (Prach et al. 2024) Prach, Bernd, and Christoph H. Lampert. "Intriguing Properties of Robust Classification." arXiv preprint arXiv:2412.04245 (2024).

---

### Decision · Program_Chairs · 2025-05-01

**Decision:**

Accept (poster)

**Comment:**

The paper received the following ratings: Weak accept, Accept, Weak reject, Accept. This paper proposes a scalable method for constructing orthogonal convolutions, combining BCOP and RKO methods. A parallel computing technique is proposed to improve performance significantly and supports additional functionalities such as stride and dilation. Experimental results showcase its effectiveness, including reduced parameter and computational overhead. Multiple weaknesses were also raised, including overstated novelty, limited theoretical guarantees, and scalability concerns. After the author-reviewer discussion, most reviewers agree that its practicality, good implementation, and accessible presentation, making a good contribution to neural network architecture research.